# Impact of revegetation of the Loess Plateau of China on the regional growing season water balance

Jun Ge[1, 2], Andrew J. Pitman[2], Weidong Guo[1, 3], Beilei Zan[4, 5], Congbin Fu[1, 3]

[1]Institute for Climate and Global Change Research, School of Atmospheric Sciences, Nanjing University, Nanjing, 210023, China.

[2]ARC Centre of Excellence for Climate Extremes and Climate Change Research Centre, University of New South Wales, Sydney, 2052, Australia.

[3]Joint International Research Laboratory of Atmospheric and Earth System Sciences, Nanjing University, Nanjing, 210023, China.

[4]Key Laboratory of Land Surface Process and Climate Change in Cold and Arid Regions, Northwest Institute of Eco-Environment and Resources, Chinese Academy of Sciences, Lanzhou, 730000, China

[5]University of Chinese Academy of Sciences, Beijing, 100049, China

*Corresponding to*: Congbin Fu (fcb@nju.edu.cn)

**Abstract.** To resolve a series of ecological and environmental problems over the Loess Plateau, the "Grain for Green Program (GFGP)" was initiated at the end of 1990s. Following the conversion of croplands and bare land on hillslopes to forests, the Loess Plateau has displayed a significant greening trend with soil erosion being reduced. However, the GFGP has also affected the hydrology of the Loess Plateau which has raised questions whether the GFGP should be continued in the future. We investigated the impact of revegetation on the hydrology of the Loess Plateau using relatively high resolution simulations and multiple realisations with the Weather Research and Forecasting (WRF) model. Results suggest that revegetation since the launch of the GFGP has reduced runoff and soil moisture due to enhanced evapotranspiration. Further revegetation associated with the GFGP policy is likely to increase evapotranspiration further, and thereby reduce runoff and soil moisture. The increase in evapotranspiration is associated with biophysical changes, including deeper roots that deplete deep soil moisture stores. However, despite the increase in evapotranspiration our results show no impact on rainfall. Our study cautions against further revegetation over the Loess Plateau given the reduction in water available for agriculture and human settlements, without any significant compensation from rainfall.

■   1 Introduction

The Loess Plateau is a highland region of north central China, covering about 640,000 km$^2$. The loess soils are
well suited for agriculture so natural forests have been progressively converted to farmland to support the growing
population over the last 7000 years (Fu et al., 2017). However, the loess is also prone to wind and water erosion,
and the long history of deforestation is associated with soil erosion, resulting in land degradation, low agricultural
productivity and significant local poverty in some farming communities (Bryan et al., 2018; Chen et al., 2015; Fu
et al., 2017). The soil erosion aggravates the flux of sediment into the Yellow River (Fu et al., 2017; Miao et al.,
2010; Peng et al., 2010) increasing the risk of catastrophic flooding in some densely populated regions
downstream (Bryan et al., 2018; Chen et al., 2015; Fu et al., 2017).
To minimise soil erosion, mitigate flood risk, store carbon and improve livelihoods over the Loess Plateau, the
"Grain for Green Program (GFGP)" was initiated by reforesting hillslopes in the late 1990s (Bryan et al., 2018;
Fu et al., 2017; Liu et al., 2008). Consequently, the Loess Plateau has displayed a significant "greening" trend
(Chen et al. 2015; Fu et al., 2017; Li et al., 2017). The large scale vegetation restoration program has also reduced
soil erosion over the Loess Plateau and alleviated sediment transport into the Yellow River (Fu et al., 2017; Liang
et al., 2015; Miao et al., 2010; Peng et al., 2010; Wang et al., 2016).
As a consequence of the beneficial outcomes of the GFGP, further investment is planned with a commitment of
around $US33.9 billion by China through to 2050 (Feng et al., 2016). However, further revegetation over the
Loess Plateau is controversial (Cao et al., 2011; Chen et al., 2015; Fu et al., 2017) with evidence from field (Jia
et al., 2017; Jin et al., 2011; Wang et al., 2012) and satellite (Feng et al., 2017; Lv et al., 2019a; Xiao, 2014)
observations that revegetation has affected the hydrological balance of the region. Compared with croplands or
barren surfaces, the planted forests enable higher evapotranspiration associated with a larger leaf area, higher
aerodynamic roughness and deeper roots (Anderson et al., 2011; Bonan, 2008; Bright et al., 2015). Consequently,
revegetation tends to decrease soil moisture and runoff with the associated risk of limiting water availability for
agriculture, human consumption and industry (Cao et al., 2011; Chen et al., 2015; Fu et al., 2017). Indeed, the
present vegetation over the Loess Plateau, which to some extent reflects decades of reafforestation, may already
exceed the limit that the local water supply can support, and hence further revegetation may not be sustainable
(Feng et al., 2016; Zhang et al., 2018).
Despite the increasing observational evidence demonstrating that revegetation tends to impair the hydrological
balance of the Loess Plateau, the response of rainfall to revegetation over this region has commonly been
overlooked. This is mainly due to the difficulty in detecting the impact of revegetation on rainfall from
observations. As an important component of hydrological cycle of the Loess Plateau, rainfall not only controls
the terrestrial water budget, but also influences soil erosion and the discharge of sediment into the Yellow River
(Liang et al., 2015; Miao et al., 2010; Peng et al., 2010; Wang et al., 2016). Therefore, how rainfall responds to
revegetation is critical to a comprehensive assessment of the impact of revegetation on the hydrology of the region.
Indeed, if rainfall responds to revegetation, this may influence national policies on whether to continue large scale
vegetation restoration programs. Afforestation or deforestation does have the potential to affect rainfall via
changes in biogeophysical processes, but any impact of afforestation or deforestation on rainfall tends to be highly
regionally specific (Findell et al., 2006; Lorenz et al., 2016; Winckler et al., 2017).
In contrast with observations, modeling can help disentangle the impact of revegetation on rainfall from the impact
of other drivers. Cao et al. (2017) and Li et al. (2018) performed numerical experiments over the whole China and
demonstrated that the revegetation over the Loess Plateau can enhance the rainfall locally. Very recently, Lv et al.
(2019b) and Cao et al. (2019) performed simulations focussed on the Loess Plateau to examine the impact of
revegetation or afforestation on rainfall. Lv et al. (2019) reported a significant increase in rainfall while Cao et al.
(2019) found spatially divergent changes of rainfall. We also note some earlier studies investigating the response
of rainfall to land cover change across China (e.g., Chen et al., 2017; Ma et al., 2013; Wang et al., 2014).
Unfortunately, these studies either focused less on the Loess Plateau (Ma et al., 2013) or applied land cover
changes unable to reflect the revegetation of the Loess Plateau (Chen et al., 2017; Wang et al., 2014). Therefore,
large uncertainties remain in the response of rainfall to revegetation of the Loess Plateau owing to inconsistent
conclusions derived from limited studies. We note Li et al. (2018) reported that the increased rainfall due to
revegetation over North China (covering but not limited to the Loess Plateau) was large enough to compensate
for the increase in evapotranspiration and resulted in little impact on soil moisture. This simulated negligible soil
moisture change associated with revegetation is contradicted by extensive studies based on observations (e.g.,
Feng et al., 2017; Jia et al., 2017; Wang et al., 2012). Here, we note it might be unfair to directly compare the
observational and modeling results because observational results commonly incorporate multiple factors and
modeling results are subject to uncertainties in both land cover change and biophysical parametrization schemes
implemented in models (de Noblet-Ducoudre et al. 2012; Pitman et al. 2009). These intrinsic differences between
observational and modeling cannot fully account for the disagreement on the runoff and soil moisture change due
to revegetation over the Loess Plateau. Thus, the impact of revegetation on the hydrology of the Loess Plateau
remains unclear and needs careful re-evaluations.
In this study, we examine the impact of revegetation following the launch of the GFGP on the hydrology of the
Loess Plateau using relatively high resolution simulations with the Weather Research and Forecasting model. We
also examine the impact of further revegetation on the hydrology of the Loess Plateau with the goal of providing
helpful information to policymakers. As far as we know, there has been no study investigating how the regional
hydrology would be affected by further revegetation over the Loess Plateau, something important for informing
policymakers on the mitigation and adaptation of climate change for this region. Additionally, the vegetation over
the Loess Plateau is fragile and highly dependent on the water availability (Fu et al. 2017). How the hydrology
would be impacted by further revegetation determines the water availability, and in turn how much more
revegetation can be sustained over the Loess Plateau. Neglecting this process risks errors in assessing the upper
threshold of vegetation of the Loess Plateau (Feng et al., 2016; Zhang et al., 2018). Given the importance of
revegetation over the Loess Plateau now and in the future we examine the impact of further revegetation on the
hydrology of the Loess Plateau and pay particular attention to the response of rainfall to revegetation.

■   **2 Methods**
■   **2.1 Model configuration**
The Weather Research and Forecasting (WRF, version 3.9.1.1, Skamarock et al., 2008), a fully coupled land-
atmosphere regional weather and climate model, was used in our study. WRF has been shown to perform well in
dynamic downscaling of regional climate over China (e.g., He et al., 2017; Sato and Xue, 2013; Yu et al., 2015).
Additionally, WRF has been used to study the impact of land use and land cover change on the hydrological
balance at regional scales (Deng et al., 2015; Zhang et al., 2018). While WRF is therefore potentially suitable for
evaluating the impact of revegetation on the hydrology of the Loess Plateau we undertake an evaluation of WRF
in simulating surface air temperature and rainfall for this region (See Section 3.1). To perform simulations at high
spatial resolution over the Loess Plateau region, we applied two-way nested runs, with two domains at different
grid resolutions running simultaneously. The ERA-Interim reanalysis data (Dee et al., 2001, Table 1) provided
the boundary conditions for the larger and coarser resolution (30 km) domain, and the larger domain provided
boundary conditions for the smaller and higher resolution (10 km) domain. The ERA-Interim reanalysis data also
provided the initial conditions for both domains. Using a lambert projection, the larger domain was centred at
100°E, 37°N, with 180 grid points in west-east direction and 155 grid points in south-north direction, covering
most of China and some surrounding regions (Fig. 1a). The inner domain covers the entire Loess Plateau with 166
grid points in west-east direction and 151 grid points in south-north direction (Fig. 1a and 1b). Both domains had
28 sigma levels in vertical direction with the top level set at 70 hPa. Fig. 1b shows the region analysed in this
paper.
The main physical parameterization schemes used in our study included the WRF Single-Moment 6-class scheme
(Hong and Lin, 2006) for microphysics, the Dudhia scheme (Dudhia, 1989) for shortwave radiation, the Rapid
Radiative Transfer Model (RRTM, Mlawer et al., 1997) for longwave radiation, a revised MM5 scheme (Jimenez
et al., 2012) for the surface layer, the Noah Land Surface Model (Ek, 2003), the Yonsei University scheme (Hong
et al., 2006) for the planetary boundary layer, and the Kain-Fritsch scheme (Kain, 2004) for cumulus convection.
The Noah Land Surface Model used the Unified NCEP/NCAR/AFWA scheme with soil temperature and moisture
in four layers (1st layer: 0-10 cm, 2nd layer: 10-40 cm, 3rd layer: 40-100 cm, 4th layer: 100-200 cm), fractional snow
cover and frozen soil physics. A sub-tiling option considering three land cover types within each grid cell was
applied to help improve the simulations of the land surface fluxes and temperature (Li et al., 2013).
■   **2.2 Data**
■   **2.2.1 Satellite data**
We used satellite observed land cover type obtained from the Moderate Resolution Imaging Spectroradiometer
(MODIS) Land Cover Type product (MCD12Q1, Version 6, Friedl and Sulla-Menashe, 2019, Table1). This
provides land cover types based on International Geosphere-Biosphere Program (IGBP) classification scheme
(Table 2) globally at a spatial resolution of 500 m, and at yearly intervals from 2001 to 2017. The MCD12Q1
Version 6 is improved over previous versions via substantial improvements to algorithms, classification schemes
and spatial resolution (Sulla-Menashe et al., 2019). We changed the land cover type within the Loess Plateau
while retaining the default land cover type for other regions in our experiments (see details in Section 2.3).
Therefore, the MCD12Q1 data were reprojected to Geographic Grid data with a resolution of 30 second
(approximately 0.9 km) by the MODIS Reprojection Tool to make them consistent with the default land cover
map in WRF.
Key land surface biogeophysical parameters include the green vegetation fraction (*VEGFRA*), snow free albedo
($\alpha$), leaf area index (*LAI*), and the background roughness length ($Z_0$). The fraction of Photosynthetically Active
Radiation (*FPAR*) can be used as a proxy of *VEGFRA* (Kumar et al., 2014; Liu et al., 2006) enabling both *VEGFRA*
and *LAI* data to be obtained from the MODIS Terra+Aqua LAI/FPAR product (MCD15A2H, Version 6, Myneni
et al., 2015a, Table 1). This provides 8-day composite *LAI* and *FPAR* globally at a spatial resolution of 500 m
since 4[th] July, 2002. The MODIS Terra LAI/FPAR product (MOD15A2H, Version 6, Myneni et al., 2015b, Table
1) was also used to provide observations prior to 2002 as it started on 8[th] February, 2000. Although MOD15A2H
has a longer span time, MCD15A2H is generally preferred. This is because only observations from the MODIS
sensor on NASA's Terra satellite is used to generate MOD15A2H, but observations from sensors on both Terra
and Aqua satellites are used for MCD15A2H. The MCD15A2H and MOD15A2H Sinusoidal Tile Grid data were
reprojected before use. The 8-day *LAI* and *FPAR* data were composited to monthly data to make them suitable for
WRF.
As we only focus on the growing season (see Section 2.3.1), $\alpha$ can be assumed to be equivalent to satellite observed
snow-free albedo. The $\alpha$ data was derived from the blue sky albedo for shortwave provided by the Global Land
Surface Satellite (GLASS) product (Liang and Liu, 2012, Table 1). This provides an 8-day composite albedo
globally at a spatial resolution of 0.05° from 1981 to present. Compared with the MODIS albedo product, the
GLASS albedo product has a higher temporal resolution and captures the surface albedo variations better (Liu et
al. 2013). The 8-day $\alpha$ data were composited to monthly data.
The background roughness length ($Z_0$) was calculated following Eq. (1):
$$Z_0 = Z_{min} + \frac{VEGFRA - VEGFRA_{min}}{VEGFRA_{max} - VEGFRA_{min}} \times (Z_{max} - Z_{min}) \tag{1}$$
where $Z_{max}$ and $Z_{min}$ were land cover dependent maximum and minimum background roughness length
respectively, provided by lookup tables. *VEGFRA*, *VEGFRA*$_{max}$ and *VEGFRA*$_{min}$ are the instantaneous, maximum
and minimum green vegetation fraction, which were calculated from satellite observed *VEGFRA* (equal to *FPAR*)
which would be implemented in WRF (see Section 2.3).
■ **2.2.2 Observation data**
To evaluate the WRF model performance in simulating the surface air temperature and rainfall over the Loess
Plateau, we used a gridded observation dataset developed by the National Meteorological Information Centre of
the China Meteorological Administration (Zhao et al., 2014, Table 1). The dataset provides monthly surface air
temperature and rainfall at a spatial resolution of 0.5° from 1961 to present and was produced by merging more
than 2400 station observations across China using Thin Plate Spline interpolation. The dataset has been widely
used to analyse the surface air temperature and rainfall over the Loess Plateau (Sun et al., 2015; Tang et al., 2018).
To facilitate the comparison between simulations and observations, the observation data were bilinearly
interpolated to the WRF inner domain grid.
■ **2.3 Experiment design**
■ **2.3.1 The impact of revegetation since the launch of the GFGP**
To examine the impact of revegetation on the hydrology of the Loess Plateau since the launch of the GFGP we
conducted a control experiment ($LC_{2001}$) and a sensitivity experiment ($LC_{2015}$). For the $LC_{2001}$, satellite observed
land cover type, *VEGFRA*, *LAI* and $\alpha$ in 2001 were used to approximate land cover type and land surface
biogeophysical parameters before the launch of the GFGP. There is a one-year gap between the launch of the
GFGP (end of 1999) and 2001, but any bias introduced by this gap is small compared with the changes in land
cover type and land surface biogeophysical parameters between 1999 and present. Satellite observed land cover
type, *VEGFRA*, *LAI* and $\alpha$ in 2015, representing the current land cover type and land surface biogeophysical status,
were used for the $LC_{2015}$. Model configurations were identical for the $LC_{2001}$ and $LC_{2015}$ except for land cover type
and land surface biogeophysical parameters. Comparing the $LC_{2001}$ and $LC_{2015}$ therefore isolates the impact of
revegetation since the launch of the GFGP.
We note that the difference between $LC_{2001}$ and $LC_{2015}$ should not be regarded as equivalent to the impact of GFGP
for two reasons. First, actual changes in land cover type since the launch of the GFGP are highly spatially
heterogeneous due to various anthropogenic activities including GFGP, irrigation and urbanization. MCD12Q1
suggests that most changes in land cover type have occurred in the south Loess Plateau (SLP, 105-111°E, 35-
37°N) and east Loess Plateau (ELP, 111-114°E, 35-39°N) (Fig. 2a, 2c, 2e and 2g). In addition to the gain of forests
(including evergreen needleleaf, evergreen broadleaf, deciduous needleleaf, deciduous broadleaf and mixed
forests) and savannas (including woody savannas and savannas), other changes in land cover type include the
expansion of croplands (including croplands and cropland/natural vegetation mosaics) at the expense of grasslands
and savannas (Fig. 2g). These increased croplands revealed by the MODIS land cover product, which seem
unlikely, have been reported previously (Fan et al., 2015; Lv et al., 2019), and are likely associated with expanded
irrigation activities along the Yellow River (Fan et al., 2015; Zhai et al., 2015). Second, the observed *VEGFRA*,
*LAI* and $\alpha$ changes also incorporate other factors including improved agricultural management, climate variability,
rising atmospheric $CO_2$ concentration and nitrogen deposition (Li et al., 2017; Fan et al., 2015; Piao et al., 2015).
As shown in Fig. 3a, 3c, 3e, and 3g, the biogeophysical changes are not strictly limited to the regions undergoing
changes in land cover type. For example, the $\alpha$ decrease mostly occurs over grasslands in northwest (Fig. 3e),
where land cover type is rarely changed (Fig 2c). This decreased $\alpha$ is attributed to increased precipitation as well
as the restoration of grasslands benefiting from the Returning Rangeland to Grassland Program launched in 2003
over this region (Zhai et al., 2015). In contrast, the $\alpha$ change is negligible in the SLP and ELP, owing to the
combined effects of increased forests (Fig. 2a) and croplands (Fig. 2d). Overall however, the MCD12Q1
demonstrates a significant greening trend (increased *VEGFRA*, *LAI* and $Z_0$ and decreased $\alpha$) over the Loess Plateau
since the launch of the GFGP (Fig. 3), which are spatially consistent with previous studies (e.g., Cao et al., 2019;
Xiao, 2014; Zhai et al., 2015).
Both $LC_{2001}$ and $LC_{2015}$ were run from 1st May to 30th September for years from 1996 to 2015 resulting in twenty
realisation members for each of $LC_{2001}$ and $LC_{2015}$. We only run for the growing season; any impact of revegetation
should be most apparent during the growing season given that over 70% of the annual rainfall occurs over the
Loess Plateau in this season (Sun et al., 2015; Tang et al., 2018).
■  **2.3.2 The impact of further revegetation on the Loess Plateau**
If the GFGP is continued in the future, further revegetation could impact the hydrology of the Loess Plateau. We
therefore conducted a third experiment ($LC_{futr}$) in which the coverage of forests was assumed to be maximum over
the Loess Plateau following the policy of the GFGP. To maximise forests we first assumed all croplands and
barren on hillslopes were converted to forests. Second, we assumed savannas or forests with low coverage (e.g.,
low *VEGFRA*) became dense forests. The land cover and land surface biogeophysical parameters for the $LC_{futr}$
were then constructed following two steps.
First, all croplands, barren and savannas pixels on hillslopes (>15°) were replaced by forests pixels over the Loess
Plateau based on the land cover map of 2015. The slope is derived from the Shuttle Radar Topography Mission
(SRTM version 2.0, Table 1) Digital Elevation Model at a spatial resolution of 3 second (about 90 m). The pixel
resolution of the land cover type is 30 second, so every land cover type pixel covered 100 ($10 \times 10$) slope values.
To maximise the revegetation, land cover type pixels with maximum slope values over 15° were regarded as
hillslopes. For a pixel to be changed, the forest class was determined by the class of neighbouring forests pixels,
considering the adaptation of planted trees to local climate. Using this strategy, forests pixels increased by 164%
and croplands pixels decreased by nearly a half in the constructed land cover map compared with the land cover
type in 2001, with most conversions occurring in SLP (Fig. 2b and 2h).
Second, we constructed the *VEGFRA*, *LAI* and *α* map in line with the land cover type constructed in the first step.
For each forests class, we screened out the "dense forests" pixels with *VEGFRA* over the 95th percentile among
the pixels labelled as the same forests class over the Loess Plateau. The monthly values of *VEGFRA*, *LAI* and *α*
of the "dense forest" pixels were calculated for each forests class. We then adjusted the monthly *VEGFRA*, *LAI*
and *α* of other "non-dense forests" pixels to the values of the "dense forests" pixels. Using this strategy, all forests
pixels over the Loess Plateau were changed to more dense forest. Consequently, the Loess Plateau shows an
amplified greening trend in $LC_{futr}$, especially in SLP (Fig. 3b, 3d, 3f and 3h).
The $LC_{futr}$ was run from 1st May to 30th September for years from 1996 to 2015. Therefore comparing $LC_{2001}$ and
$LC_{futr}$ isolates the impact of further revegetation on the hydrology of the Loess Plateau.
■    **2.3.3 Identification of the impact of revegetation**
Model internal variability is defined as the difference between realisation members where the only differences are
the initial conditions. These differences result from nonlinearities in the model physics and dynamics (Giorgi and
Bi, 2000; Christensen et al., 2001). This means some differences between $LC_{2001}$ and $LC_{2015}$ (or $LC_{futr}$) will be
caused by internal variability in addition to revegetation (Lorenz et al., 2016; Ge et al., 2019). To minimise the
impact of internal model variability we performed multiple simulations for the year 2001 by changing initial
conditions. Specifically, we carried out a pair of experiments named $LCENS_{2001}$ and $LCENS_{2015}$, which were the
same as $LC_{2001}$ and $LC_{2015}$ except that $LCENS_{2001}$ and $LCENS_{2015}$ were only run for the year 2001 but initialized
for each day between 21st to 30th April, and ending on 30th September. This led to a total of eleven members
(including the members with initial dates of 1st May in $LC_{2001}$ and $LC_{2015}$) for $LCENS_{2001}$ and $LCENS_{2015}$
respectively. Comparing $LCENS_{2001}$ and $LCENS_{2015}$, simulated changes were likely robust if the impact from
revegetation was large and consistent relative to the differences caused by the change in the initial condition.
Results before 1st June was discarded as spin-up time in each simulation. Our analysis focusses on June, July,
August and September (JJAS) averages.

■   **2.5 Local significance test**

To test the statistical significance of the local impact of revegetation on the hydrology we calculate a grid-point
by grid-point Student's *t*-test. This tests the null hypothesis that the two groups of data are from independent
random samples from normal distributions with equal means and equal but unknown variances. The local
difference is regarded as statistically significant when the *p*-value of the two-tailed *t*-test passes the significance
level of 95%.

■   **3 Results**

■   **3.1 Evaluation of WRF's skill in simulating temperature and rainfall**

We first evaluate WRF's simulation of surface 2m air temperature (*T2*) and rainfall (*RAIN*), the quantities with
the most credible observations available over the Loess Plateau, by comparing the averaged value of the eleven
members in $LCENS_{2001}$ with the observed values in 2001. After topographic correction (Zhao et al., 2008), WRF
simulates *T2* over the Loess Plateau mostly within 2°C of the observations (Fig. 4a, 4c, 4e) although there are
small areas where WRF simulates warmer temperatures than the observations by 4°C. The model also performs
well in simulating *RAIN* (Fig. 4b, 4d, 4f) including a region of higher observed rainfall from the southwest to the
central Loess Plateau. The *RAIN* bias between the WRF simulations and the observations is below 0.5 mm/day
for almost the entire Loess Plateau (Fig. 4f). Larger *RAIN* biases mostly occur around the eastern and southern
borders of the Loess Plateau, most likely due to extremely complex topography in these locations. Since we focus
on the impact of land cover change on the hydrology of the region, the reasonable simulation of *RAIN* gives us
confidence in the results from WRF, particularly in SLP.
■    **3.2 Impacts on surface fluxes**
We first examine the change in the land surface radiation budget, energy and water fluxes as these are directly
impacted by changes in land cover type and the surface biogeophysical parameters. Comparing $LC_{2001}$ and $LC_{2015}$
($LC_{2015}$-$LC_{2001}$), land surface net radiation ($R_{net}$), latent heat flux ($Q_E$) and sensible heat flux ($Q_H$) changes mainly
occur where land cover type and land surface biogeophysical parameters are changed, suggesting a strong local
effect on $R_{net}$, $Q_E$ and $Q_H$. $R_{net}$ increases by around 5-20 $W \cdot m^{-2}$ (Fig. 5a), over most of the region due to a reduction
in $\alpha$ (Fig. 3e). While $Q_E$ increases by 10-30 $W \cdot m^{-2}$ (Fig. 5c) and $Q_H$ reduces by around 10 $W \cdot m^{-2}$ (Fig. 5e), mostly
in SLP and ELP as a result of increased *VEGFRA*, *LAI* and $Z_0$ (Fig. 3a, 3c and 3g). Changes in $R_{net}$ and $Q_E$ are
statistically significant at a 95% confidence level over most of the region, but statistically significant changes in
$Q_H$ are mostly limited to SLP and ELP (see the embedded subplots in each panel, Fig. 5a, 5c and 5e). As a
consequence of further revegetation ($LC_{futr}$-$LC_{2001}$), $R_{net}$, $Q_E$ and $Q_H$ changes are intensified (Fig. 5b, 5d and 5f),
especially in SLP where large areas of croplands are converted to forest leading to large changes in land surface
biogeophysical parameters in $LC_{futr}$ (Fig. 2 and 3).
Focusing on SLP, the increase in evapotranspiration (*ET*) is 0.49 $mm \cdot day^{-1}$ between $LC_{2001}$ and $LC_{2015}$ (Fig. 6a).
WRF simulates further water loss (0.85 $mm \cdot day^{-1}$) through *ET* if the revegetation is continued in the future (Fig.
6c). For ELP, where relative fewer croplands or barren can be further converted to forests in $LC_{futr}$, the future *ET*
increase is still considerable (0.72 $mm \cdot day^{-1}$, Fig. 6b and 6d). The values of regional mean *ET* change among the
twenty members of $LC_{2015}$-$LC_{2001}$ and $LC_{futr}$-$LC_{2001}$ remain consistently positive over SLP and ELP. This indicates
that the simulated higher *ET* is a consistent result from WRF as a consequence of the revegetation since the launch
of the GFGP, and is likely to be further strengthened by continued revegetation over the Loess Plateau.
■    **3.3 Impacts on rainfall**
Increased *ET* can contribute to the formation of clouds and rainfall, and we therefore examine whether this is the
case for the Loess Plateau. The *RAIN* is composed of convective rainfall (*RAINC*) calculated by the cumulus
convection scheme, and non-convective rainfall (*RAINNC*) calculated by microphysics scheme in WRF. Thus we
separate *RAINC* and *RAINNC* changes in addition to the *RAIN* change in Fig.7. As for $LC_{2015}$-$LC_{2001}$, the change
in *RAIN* is spatially heterogeneous, with an increase of up to 1.2 mm·day$^{-1}$ in small parts of the northeast and a
decrease around -1.0 mm·day$^{-1}$ along the southeast border of the Loess Plateau (Fig. 7a). The *RAIN* change is
divided almost evenly between *RAINC* and *RAINNC* (Fig. 7c and 7e). However, most of the *RAIN*, *RAINC* and
*RAINNC* changes are not statistically significant. In terms of LC$_{futr}$-LC$_{2001}$, *RAIN*, *RAINC* and *RAINNC* are not
significantly changed by further revegetation (Fig. 7b, 7d and 7f). Moreover, the increased *RAIN* in northeast
Loess Plateau occurring in LC$_{2015}$-LC$_{2001}$ dissipate when further revegetation is implemented while the changes
in both land cover type and biophysical parameters are relatively small over this regions. This increased *RAIN*
should be maintained in LC$_{futr}$-LC$_{2001}$ if the change in *RAIN* is robust for LC$_{2015}$-LC$_{2001}$. We will analyse the
increased *RAIN* of the northeast Loess Plateau in LC$_{2015}$-LC$_{2001}$in Section 3.6.
For both LC$_{2015}$-LC$_{2001}$ and LC$_{futr}$-LC$_{2001}$ cases, most *RAIN* changes seem to be randomly scattered around the
Loess Plateau instead of being located coincident with SLP or ELP where land cover type, land surface
biogeophysical parameters and land surface fluxes are most strongly modified (Fig. 7a and 7b). In contrast, the
*RAIN* change is negligible over SLP and ELP for both LC$_{2015}$-LC$_{2001}$ and LC$_{futr}$-LC$_{2001}$ cases (Fig. 6 and 7).
However, the *RAIN* change in individual realisation is not small, e.g., the *RAIN* change varies from -2.11 to 2.21
mm·day$^{-1}$ over the ELP for LC$_{2015}$-LC$_{2001}$ (Fig. 6b). So averaging the divergent *RAIN* changes among the twenty
members causes a negligible *RAIN* change overall. This large variability in *RAIN* changes among the twenty
members can be attributed to either different boundary conditions (background climate), which causes the impact
of land cover change to diverge (Pitman et al., 2011), or model internal variability. This will be further analysed
in Section 3.6.
■   **3.4 Impacts on runoff**
As a consequence of the significant *ET* increase and negligible and statistically insignificant *RAIN* change,
underground runoff (*UDROFF*) is reduced by up to 1.5 mm·day$^{-1}$ locally for LC$_{2015}$-LC$_{2001}$ (Fig 8c). Averaged
over the SLP and ELP, the *UDROFF* decreases by 0.16 mm·day$^{-1}$ (-23%) and 0.34 mm·day$^{-1}$ (-23%) for SLP and
ELP respectively (Fig. 6a and 6b). These *UDROFF* changes are not statistically significant and vary strongly
among the twenty members, suggesting a large uncertainty in the *UDROFF* change. WRF simulated a larger
*UDROFF* decrease due to further revegetation (Fig. 8d), especially over SLP and ELP where the regional mean
*UDROFF* decreases by 0.38 mm·day$^{-1}$ (-54%) and 0.63 (-42%) respectively (Fig. 6c and 6d). These *UDROFF*
decreases are statistically significant at a 95% confidence level for both SLP and ELP. Moreover, the upper
quartile of *UDROFF* changes among the twenty members systematically shift below the 0 mm·day$^{-1}$ value for
both the SLP and ELP. These results indicate a larger chance of the *UDROFF* decrease if the revegetation is
continued over the SLP and ELP. Moreover, the spatial change in *UDROFF* is consistent with that of the net
budget of *RAIN* and *ET* (*RAIN-ET*) for both LC$_{2015}$-LC$_{2001}$ and LC$_{futr}$-LC$_{2001}$ (Fig. 8e and 8f), suggesting that the
*UDROFF* change can be mostly explained by the change of *RAIN-ET*. We also note some *UDROFF* changes in
adjacent regions of the Loess Plateau (Fig. 8c and 8d) associated with *RAIN* changes (Fig. 7a and 7b).
Compared with the *UDROFF* change, the surface runoff (*SUROFF*) change are mostly small for both LC$_{2015}$-
LC$_{2001}$ and LC$_{futr}$-LC$_{2001}$ (Fig. 8a and 8b). However, the relative change of *SUROFF* is considerable, especially
for the LC$_{futr}$-LC$_{2001}$ case in which *SUROFF* decreased by 21% for the SLP and 14% for the ELP respectively
(Fig. 6c and 6d). We also find the upper quartile of the *SUROFF* change systematically shifts below the 0 mm·day$^{-}$
$^{1}$ value although the *SUROFF* change are not statistically significant for the LC$_{futr}$-LC$_{2001}$.
■   **3.5 Impacts on soil moisture**
In addition to the decline in runoff, the soil moisture (*SMOIS*) of each layer is significantly reduced over the Loess
Plateau for LC$_{2015}$-LC$_{2001}$ (Fig. 9a, 9c, 9e and 9g) with larger decreases in the middle two layers. The regional
mean *SMOIS* for the SLP decreases by 0.02 m·m$^{-3}$ (-8%) and 0.03 m·m$^{-3}$ (-12%) for the second and third layers
(Fig. 6a). WRF simulated further falls in soil moisture following further revegetation, with a larger impact on
deeper soil layer moisture (Fig. 9b, 9d, 9f and 9h). For example, the decrease in regional mean soil moisture of
the bottom layer for the SLP varies from -0.01 (or -5%) in LC$_{2015}$-LC$_{2001}$ (Fig. 6a) to -0.04 (or -17%) in LC$_{futr}$-
LC$_{2001}$ (Fig. 6c). Similar to the *UDROFF* change, the spatial change in *SMOIS* for each layer is consistent with
that of *RAIN-ET* for both LC$_{2015}$-LC$_{2001}$ and LC$_{futr}$-LC$_{2001}$ (Fig.8e and 8f).
■   **3.6 Robust identification of rainfall change**
We found a large variability in changes in *RAIN* among the twenty members over the SLP and ELP for both
LC$_{2015}$-LC$_{2001}$ and LC$_{futr}$-LC$_{2001}$. We next examine whether these can be attributed to revegetation. We first show
the *RAIN* change in individual members for LC$_{2015}$-LC$_{2001}$ (Fig. 10). The large variability of *RAIN* changes among
the twenty members occur throughout the study region. Even the increase in *RAIN* over the northeast Loess Plateau
(Fig. 7a), which is available by comparing multiyear mean *RAIN* between LC$_{2001}$ and LC$_{2015}$, is not consistent for
every year. As for the northeast Loess Plateau, the *RAIN* shows an increase in 8 years (1997, 2001, 2003, 2004,
2007, 2010, 2012 and 2015), decrease in 5 years (1996, 1999, 2006, 2009 and 2014) and negligible changes in
other 7 years. This results in a net increase in *RAIN* over the twenty years, but a different selection of years could
show an overall decrease (the result is similar for $LC_{futr}$-$LC_{2001}$, not shown). Similarly, other statistically significant
*RAIN* changes occur in the study region (e.g., decreased *RAIN* to the southwest Loess Plateau shown in Fig. 7a)
but these are not consistent across the twenty years. As mentioned earlier, this large variability in *RAIN* changes
among the twenty members is possibly attributed to different boundary conditions (background climate), and we
next examine whether this is true over the Loess Plateau.
We note that the pattern of *RAIN* change in 2001 is very similar to the multiyear averaged one, but with a larger
magnitude (Fig. 7a and 10f). The *RAIN* increase of the northeast Loess Plateau in just 2001 explains about 30%
of the multiyear mean *RAIN* increase in the same region. We therefore show the *RAIN* change in each realisation
for $LCENS_{2015}$-$LCENS_{2001}$ in Fig. 11. These eleven ensemble members share the same boundary conditions with
small differences in initial conditions. In contrast with the increased *RAIN* obtained from setting initial date on 1st
May (Fig. 10f), the *RAIN* changes are modified by an advance of 1 to 10 days in initial conditions. For example,
WRF cannot simulate the increased *RAIN* over northeast Loess Plateau when using an initial date of 22nd, 25th,
27th and 30th April, highlighting that the *RAIN* change is very sensitive to the initial conditions. Thus, the *RAIN*
increase in 2001 with an initial date of 1st May is likely associated with internal variability rather than revegetation.
In another words, the *RAIN* change due to revegetation is negligible relative to the *RAIN* change induced by
internal variability. We therefore conclude that the multiyear averaged *RAIN* increase over northeast Loess Plateau
for $LC_{2015}$-$LC_{2001}$ (Fig. 7a) cannot be robustly linked with revegetation.
■    **3.7 How many members do we need to get a robust signal?**
Model internal variability is inevitable when we use models to investigate the impact of land cover change on
climate. The model internal variability can be minimised as the number of individual realisations is increased to
form a larger sample to calculate any average. We therefore examine the relationship between the *RAIN* change
and the number of realisation members (Fig. 12). Focusing on the SLP and ELP, the range of *RAIN* change
decreases as the number of realisations increase. For example, the *RAIN* change over the ELP varies from -0.97
to 1.07 mm·day⁻¹ when only three members are included. The range of *RAIN* is narrowed to between -0.25 and
0.24 mm·day$^{-1}$ when fifteen members are simulated. It is similar for LCENS$_{2015}$-LCENS$_{2001}$; the range in the
change in *RAIN* decreases as the number of simulation members increases. The change in *RAIN* suggests an
increase of 0.48 and 0.40 mm·day$^{-1}$ for the SLP and ELP respectively when the simulation members are increased
to eleven.
■     **4 Discussion**
Following the launch of the GFGP by China in the late 1990s, the Loess Plateau has shown a significant greening
trend, but with simultaneous concerns about water security for agriculture and other human activities. We
investigated the impact of revegetation since the launch of the GFGP on the hydrology of the Loess Plateau using
WRF. Simulations show that the revegetation of the plateau is associated with a decrease in runoff and soil
moisture as a consequence of higher evapotranspiration and little feedback from rainfall. Our results on changes
of evapotranspiration, soil moisture and runoff are broadly consistent with both field (Jia et al., 2017; Jian et al.,
2015; Jin et al., 2011) and satellite (Feng et al., 2017; Li et al., 2016; Xiao, 2014) observations. For example, the
spatial pattern of our simulated soil moisture decline in the growing season is similar to observations from the
Advanced Microwave Scanning Radiometer on the Earth Observing System by the Japanese Aerospace
Exploration Agency (Feng et al., 2017). Although the increased evapotranspiration due to revegetation of the
Loess Plateau has been examined before (e.g., Cao et al., 2017, 2019; Li et al., 2018; Lv et al., 2019), the reduction
in runoff and soil moisture in response to revegetation of the Loess Plateau, which is consistent with observations,
has been rarely reported in modeling results previously. Moreover, our simulated weak response of rainfall to
revegetation of the Loess Plateau, which is hard to determine from observations, is useful in assessing the
hydrometeorology of this region.
We also investigated the potential future impact on the hydrology of the Loess Plateau if revegetation was
continued, which has not been assessed before but is important for both scientific communities and policymakers.
WRF suggests that further revegetation would exacerbate soil moisture and runoff declines with particularly large
effects on the underground runoff and soil moisture in deeper layers. Our simulations suggested that the potential
revegetation that could still be achieved would have larger consequences than those simulated since the launch of
the GFGP. Our results provide useful advances in our understanding of the impact of further revegetation on the
Loess Plateau. For example, both Feng et al. (2016) and Zhang et al. (2018) estimated the current vegetation over
the Loess Plateau is approaching or may have exceeded the threshold of ecological equilibrium. They omitted the
potential response of rainfall to further revegetation over the Loess Plateau when predicting future thresholds
(Feng et al., 2016; Zhang et al., 2018). Our result demonstrate that there is almost no feedback of rainfall
associated with further revegetation, supporting the approach of Feng et al. (2016) and Zhang et al. (2018) in this
specific region. That said, our approach does not attempt to incorporate changes in climate over the Loess Plateau
and so the viability of large-scale reforestation in this region is not something we attempted to assess.
We focused on the response of rainfall to revegetation over the Loess Plateau, which is probably the most uncertain
of the hydrological components. WRF shows little response of rainfall to revegetation since the launch of the
GFGP, which contradicts earlier results (Cao et al., 2017, 2019; Li et al., 2018; Lv et al., 2019). Moreover, the
rainfall is weakly affected by further revegetation despite large increase in evapotranspiration. We also
demonstrate that the rainfall change is strongly affected by internal variability and a large number of realisations
are required before any impact of revegetation on rainfall might be robustly identified. We suggest that some
previous studies (Cao et al., 2017, 2019; Lv et al., 2019) based on model simulations may have exaggerated the
impact of revegetation on rainfall over the Loess Plateau due to the lack of sufficient realisations. For example,
Cao et al. (2017, 2019) and Lv et al. (2019) used the same WRF to perform only three or five member simulations,
and concluded a significant change in rainfall caused by revegetation over the Loess Plateau. More interestingly,
Cao et al. (2017) and Cao et al. (2019) obtained different conclusions on the rainfall change over the Loess Plateau
with same WRF model. They used a broadly similar experimental design but different spatial resolution (30 km
and 10 km respectively) and simulations from 2001-2002 with three ensembles and consecutive simulation from
2000-2004 respectively. We could also demonstrate large changes in rainfall over the plateau if we chose 3-5
members but we could demonstrate either large increases or large decreases in 3-5 member averages. Returning
to Fig. 6, ET shows a highly consistent increase in response to revegetation among the 20 years, suggesting that
ET change is robustly linked with revegetation. Although changes in runoff and soil moisture also show large
variability among the 20 years, the distribution of the runoff and soil moisture changes are negative biased. More
importantly, the distribution of the runoff and soil moisture changes systematically shift towards negative values.
This suggest runoff and soil moisture changes are very likely linked with revegetation. The large variability in
runoff or soil moisture changes is induced by the large variability of rainfall. Given the tight linkage between
rainfall and runoff or soil moisture, the changes in runoff or soil moisture tends to be mistakenly represented if
the rainfall change is not robustly examined, and this requires internal model variability to be thoroughly addressed.
Our studies are also subject to some caveats. First, observations of soil moisture declines associated with
revegetation can be alleviated once trees mature (Jia et al., 2017; Jin et al., 2011). Our simulations only capture
an initial decline in runoff and soil moisture linked with the higher evapotranspiration and we note that the impact
of revegetation on the long-time trend (25 - 50 years) would be valuable. Second, we used current boundary
conditions (1996-2015) for WRF to predict the impact of further revegetation on the hydrology, which means the
boundary conditions do not change in the future in response to climate change. This suggests that we might
underestimate the impact of further revegetation in the future if future climate of the Loess Plateau suffers from
large changes in response to global warming. Third, uncertainties exist in the current land surface model used to
represent the response of vegetation to climate change in future. While using satellite observations to construct
the land surface biogeophysical parameters helps overcome some land surface parameter limitations, this approach
is obviously limited looking forward in terms of the status of future vegetation. Furthermore, we note that our
results are likely model dependent as we only used one model. Although we performed relatively high resolution
(10 km for the nested domain), the cumulus convection scheme remains necessary which is a further potential
source of uncertainty. These factors account for the discrepancy between our result and another model based study
(Li et al., 2018). Li et al (2018) found a positive rainfall feedbacks to greening and consequently small changes
in runoff and soil moisture over north China using a Global Climate Model. In contrast, we demonstrate the rainfall
change is too small to compensate for the strongly enhanced evapotranspiration, causing a reduction of runoff and
soil moisture in response to revegetation over the Loess Plateau. A large ensemble of models, each with a
reasonable number of realisations, is needed to build a model independent assessment of the impact of revegetation
but this is clearly beyond the scope of this study. Last, we investigated the impact of revegetation or greening,
rather than GFGP, on the hydrology of the Loess Plateau. Directly linking our results to the impact of GFGP on
the hydrology of the Loess Plateau should be avoided.
Overall, our results highlight how revegetation of the Loess Plateau led to increased evapotranspiration and how
as a consequence the runoff and soil moisture declined. This is consistent with the understanding of land-surface
processes and how they respond to land cover change (Bonan, 2008). Critical in this impact of revegetation on
the hydrology is what happens to rainfall. If the higher evapotranspiration increases rainfall, then revegetation has
the potential to increase soil moisture and runoff. It is very likely this would be the consequences in some regions
such as Amazonia (Lawrence and Vandecar, 2015; Perugini et al., 2017; Spracklen et al., 2018) and Sahel
(Kemena et al., 2018; Xue and Shukla, 1996; Yosef et al., 2018). However, over the Loess Plateau we find no
such result and thus the higher evapotranspiration simply leads to lower soil moisture and runoff. Additionally,
Tobella et al. (2014) reported a positive impact of trees on soil hydraulic properties influencing groundwater
recharging when termite mound is taken into account in Africa. While the termite mound is rare over the Loess
Plateau suggesting this positive impact of trees is unlikely to occur. An implication of this result is that further
revegetation, which requires water to be sustained, may not be viable. We also recognize that afforestation can
help to sequester carbon, mitigate warming and alleviate soil erosion. Therefore whether and how to implement
further revegetation should be cautiously determined with the pros and cons of afforestation being carefully
weighted for the Loess Plateau.
■    **5 Conclusions**
We evaluated how the growing season hydrology of the Loess Plateau is impacted by revegetation since the launch
of the "Grain for Green Program", and by further revegetation in the future using the WRF model. We used
satellite observations to describe key biophysical parameters including decreased albedo and increased leaf area
index and fraction of photosynthetically active radiation. The observed greening trend increased
evapotranspiration but because the impact on rainfall was negligible the underground runoff and soil moisture
both decreased. Further future revegetation enhanced evapotranspiration, but still had little impact on rainfall.
Overall therefore, revegetation over the Loess Plateau leads to higher evapotranspiration, and as a consequence
lower water availability for agriculture or other human demands. Considering the negative impact of revegetation
on runoff and soil moisture, and the lack of benefits on rainfall, we caution that further revegetation may threaten
local water security over the Loess Plateau.
*Code and data availability*. The MODIS land cover type product (MCD12Q1) and LAI/FPAR products
(MCD15A2H and MOD15A2H) are available on NASA's Land Processes Distributed Active Archive Center (LP
DAAC), https://lpdaac.usgs.gov/data/. The GLASS albedo product is available on Global land surface satellite
(GLASS) products download and service, http://glass-product.bnu.edu.cn/. The ERA-Interim reanalysis data is

available on the ECMWF Data Server, https://www.ecmwf.int/en/forecasts/datasets/reanalysis-datasets/era-interim. The gridded observation dataset is available on the National Meteorological Information Centre of the China Meteorological Administration, http://data.cma.cn/data/cdcindex.html. The code of Weather Research and Forecasting model is available on http://www2.mmm.ucar.edu/wrf/users/.

*Author contributions*. CF, JG and WG led the overall scientific questions and designed the research. JG, AJP and BZ analysed the data and wrote the manuscript. All authors contributed to the discussion of the results and to revising the manuscript.

*Competing interest*. The authors declare that they have no conflict of interest.

*Acknowledgements*. This work was supported by the Natural Science Foundation of China (41775075, 41475063) and the Australian Research Council via the Centre of Excellence for Climate Extremes (CE170100023). This work is also supported by the Jiangsu Collaborative Innovation Center for Climate Change. The model simulations were conducted on the NCI National Facility at the Australian National University, Canberra. The authors also gratefully acknowledge financial support from China Scholarship Council.

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

**Tables**
**Table 1.** Descriptions of datasets used in this study

| Variable | Dataset | Time span available | Temporal resolution | Spatial resolution |
|---|---|---|---|---|
| Land cover | MCD12Q1 | 2001-2017 | Yearly | 500 m |
| LAI/FPAR | MCD15A2H | 4[th] July, 2002 to present | 8-day | 500 m |
| LAI/FPAR | MOD15A2H | 8[th] Feburary, 2000 to present | 8-day | 500 m |
| Albedo | GLASS | 1981 to present | 8-day | 0.05º |
| Initial and boundary conditions for WRF | ERA-Interim | 1979 to present | 6 hour | 0.75º |
| Surface air temperature | National Meteorological Information Centre | 1961 to present | Monthly | 0.5º |
| Rainfall | National Meteorological Information Centre | 1961 to present | Monthly | 0.5º |
| Slope | SRTM | — | — | 3 second (about 90 m) |



**Table 2.** The International Geosphere-Biosphere Programme (IGBP) classification and class descriptions

| Name | Value | Description |
|---|---|---|
| Evergreen Needleleaf Forests | 1 | Dominated by evergreen conifer trees (canopy >2m). Tree cover >60%. |
| Evergreen Broadleaf Forests | 2 | Dominated by evergreen broadleaf and palmate trees (canopy >2m). Tree cover >60%. |
| Deciduous Needleleaf Forests | 3 | Dominated by deciduous needleleaf (larch) trees (canopy >2m). Tree cover >60%. |
| Deciduous Broadleaf Forests | 4 | Dominated by deciduous broadleaf trees (canopy >2m). Tree cover >60%. |
| Mixed Forests | 5 | Dominated by neither deciduous nor evergreen (40-60% of each) tree type (canopy >2m). Tree cover >60%. |
| Closed Shrublands | 6 | Dominated by woody perennials (1-2m height) >60% cover. |
| Open Shrublands | 7 | Dominated by woody perennials (1-2m height) 10-60% cover. |
| Woody Savannas | 8 | Tree cover 30-60% (canopy >2m). |
| Savannas | 9 | Tree cover 10-30% (canopy >2m). |
| Grasslands | 10 | Dominated by herbaceous annuals (<2m). |
| Permanent Wetlands | 11 | Permanently inundated lands with 30-60% water cover and >10% vegetated cover. |
| Croplands | 12 | At least 60% of area is cultivated cropland. |
| Urban and Built-up Lands | 13 | At least 30% impervious surface area including building materials, asphalt, and vehicles. |
| Cropland/Natural Vegetation Mosaics | 14 | Mosaics of small-scale cultivation 40-60% with natural tree, shrub, or herbaceous vegetation. |
| Permanent Snow and Ice | 15 | At least 60% of area is covered by snow and ice for at least 10 months of the year. |
| Barren | 16 | At least 60% of area is non-vegetated barren (sand, rock, soil) areas with less than 10% vegetation. |
| Water Bodies | 17 | At least 60% of area is covered by permanent water bodies. |


**Table 3.** Description of the experiment design

| Experiment | Land cover | *VEGFRC* | *LAI* | *α* | Simulation period |
|---|---|---|---|---|---|
| $LC_{2001}$ | 2001 | 2001 | 2001 | 2001 | 1st May to 30th Sep. for years from 1996 to 2015 |
| $LC_{2015}$ | 2015 | 2015 | 2015 | 2015 | 1st May to 30th Sep. for years from 1996 to 2015 |
| $LC_{futr}$ | Artifically constructed land cover and land surface biogeophysical parameters (see text) | | | | 1st May to 30th Sep. for years form 1996 to 2015 |
| $LCENS_{2001}$ | 2001 | 2001 | 2001 | 2001 | From varying initial time (from 21st April to 1st May) to 30th Sep. for the year 2001 |
| $LCENS_{2015}$ | 2015 | 2015 | 2015 | 2015 | From varying initial time (from 21st April to 1st May) to 30th Sep. for the year 2001 |


 **Figures**

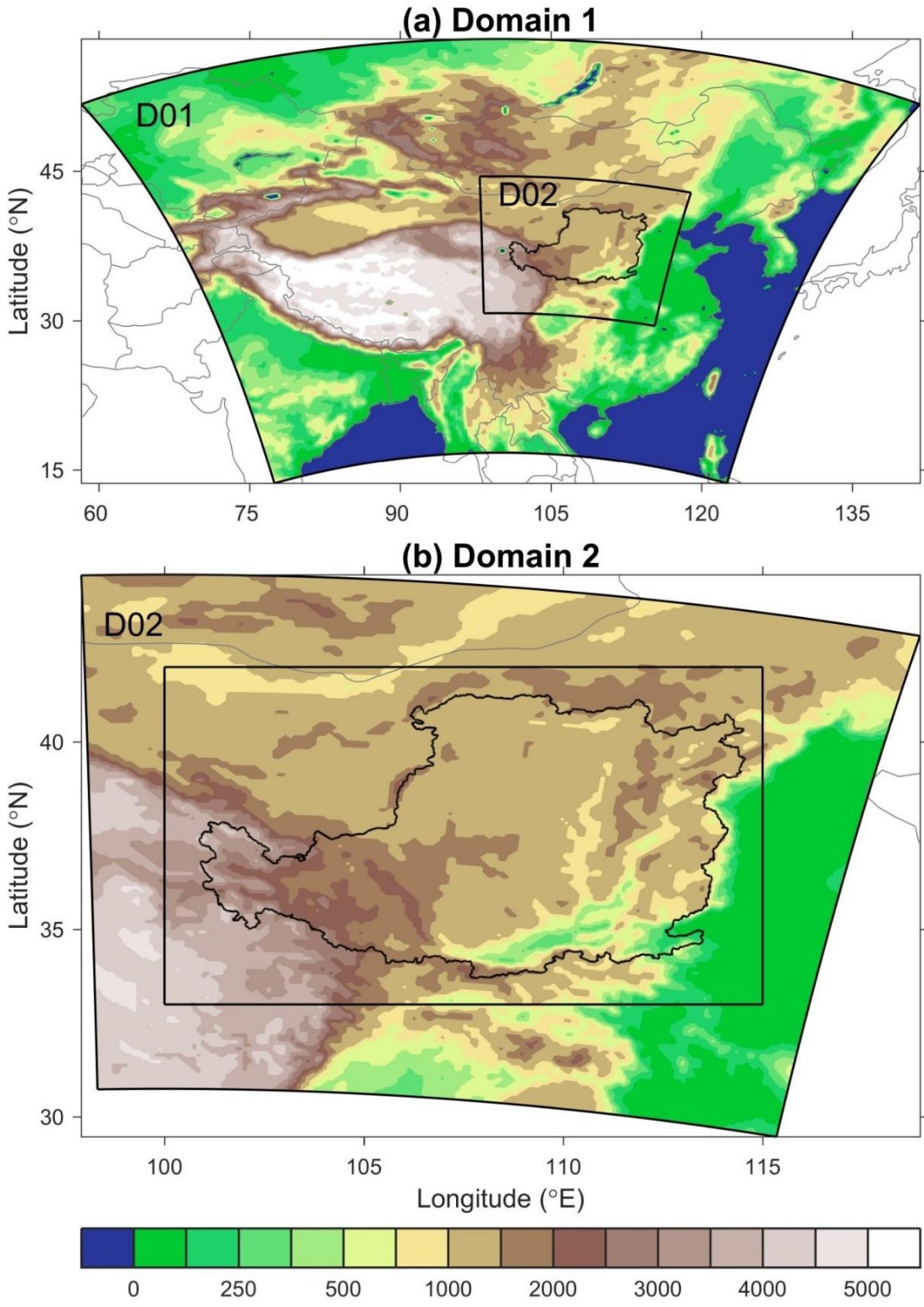


**Figure 1.** (a) The larger domain labelled D01 and (b) the inner nested domain labelled D02 configured for the
WRF model. The topography (meters above sea level) is shown as colour shading. The Loess Plateau is enclosed
by the black border. The black rectangle covers the region to be analysed in this study.

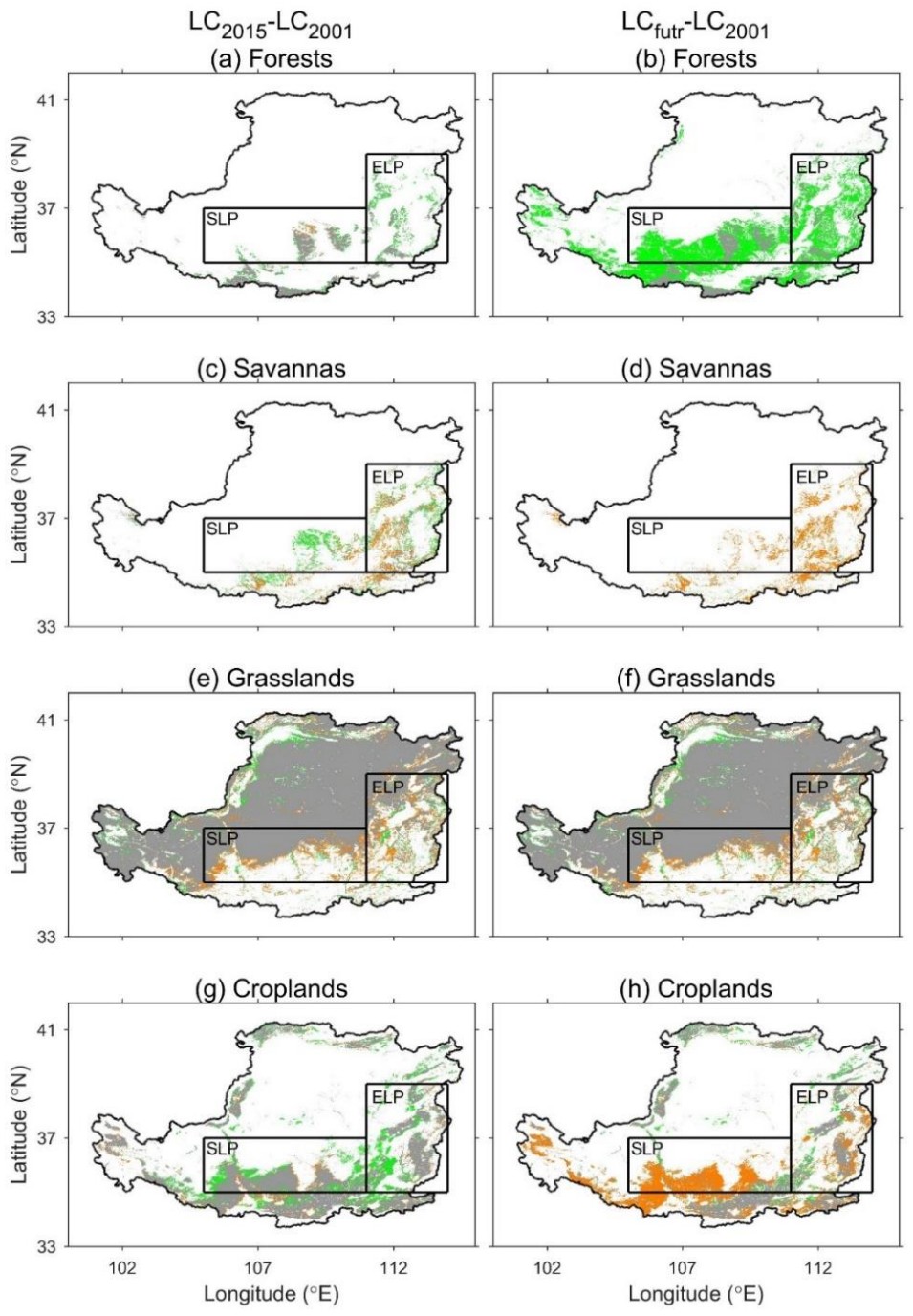


**Figure 2.** Land cover type changes (a, c, e and g) between the $LC_{2001}$ and $LC_{2015}$ ($LC_{2015}$-$LC_{2001}$), and (b, d, f and

h) between the $LC_{2001}$ and $LC_{futr}$ ($LC_{futr}$-$LC_{2001}$). The green, brown and grey colours denote the gained, lost and

unchanged land cover type respectively in the $LC_{2015}$ (a, c, e and g) and $LC_{futr}$ (b, d, f and h) compared with the

$LC_{2001}$. Forests include evergreen needleleaf, evergreen broadleaf, deciduous needleleaf, deciduous broadleaf and

mixed forests (see Table 2). Savannas include woody savannas and savannas. Croplands include croplands and

cropland/natural vegetation mosaics. The south (105-111°E, 35-37°N) and east (111-114°E, 35-39°N) Loess

Plateau are enclosed by black rectangles and labelled SLP and ELP respectively.


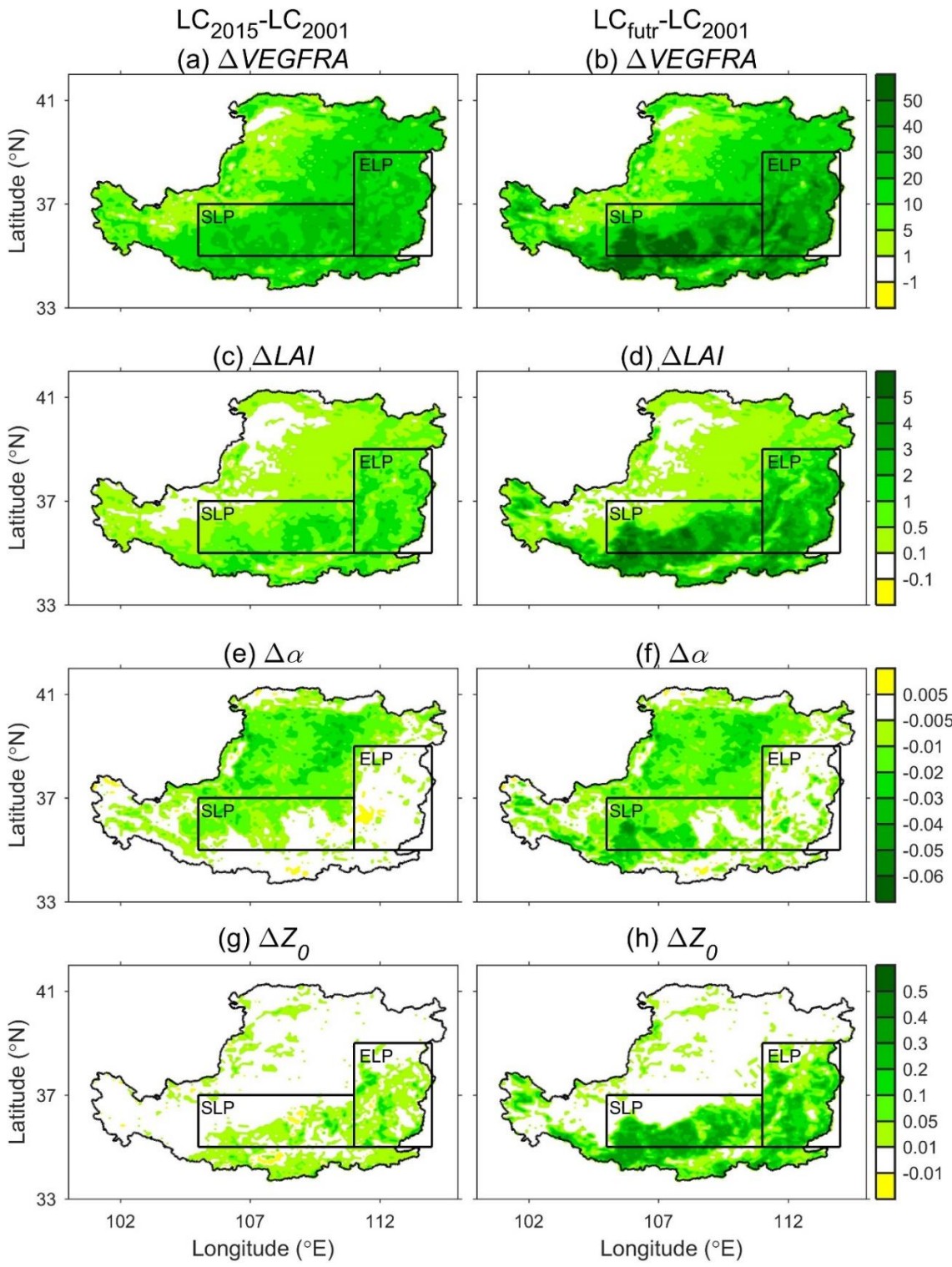


**Figure 3.** Changes in June-July-August-September mean (a and b) green vegetation fraction (%), (c and d) leaf
area index ($m^3 \cdot m^{-3}$), (e and f) albedo and (g and h) roughness length (m) between the $LC_{2001}$ and $LC_{2015}$ ($LC_{2015}$-
$LC_{2001}$; a, c, e and g), and between the $LC_{2001}$ and $LC_{futr}$ ($LC_{futr}$-$LC_{2001}$; b, d, f and h). The south (SLP) and east
(ELP) Loess Plateau regions are defined in Figure 2.

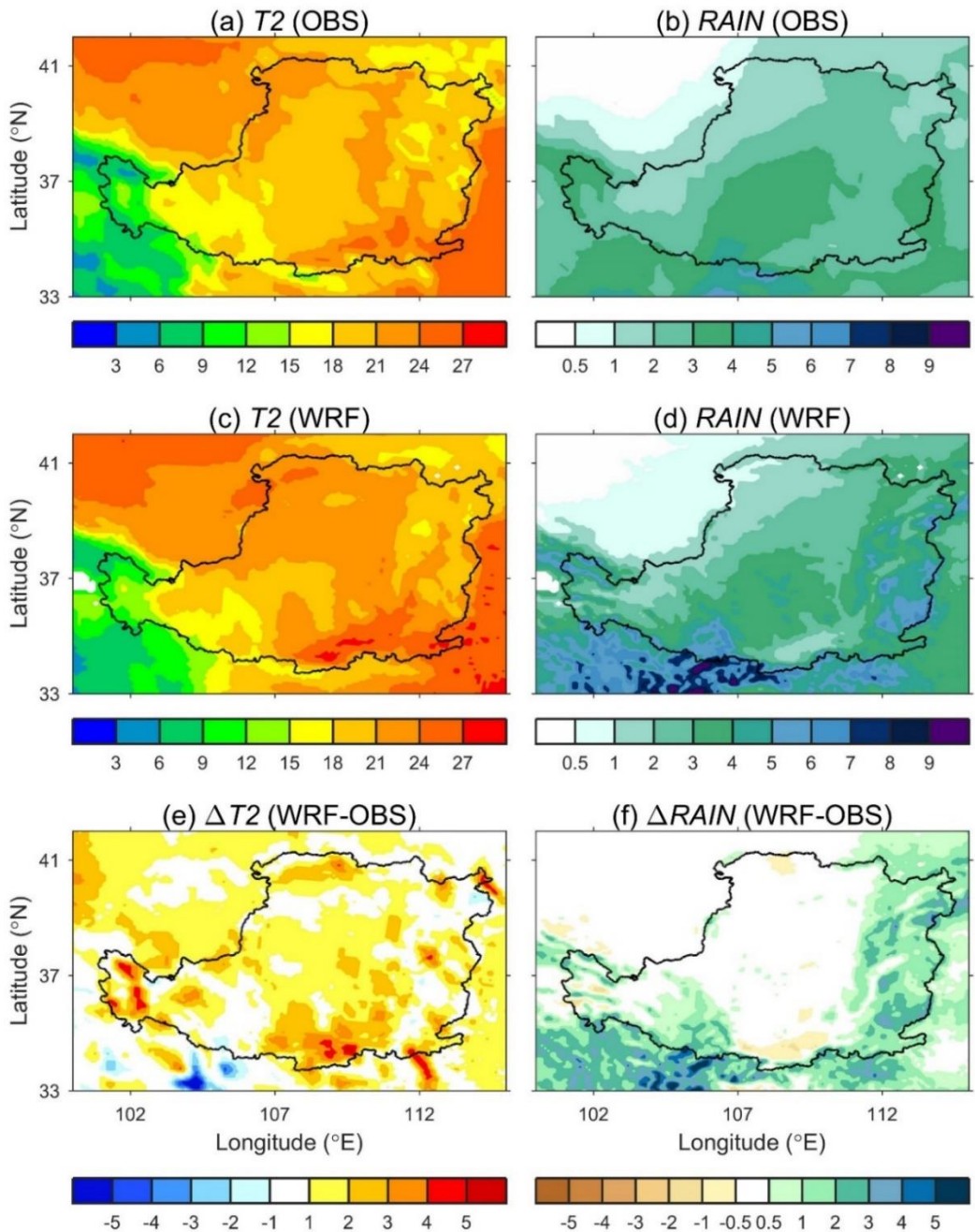


**Figure 4.** June-July-August-September (JJAS) mean (a) observed surface air temperature (ºC), (b) observed
rainfall (mm·day$^{-1}$), (c) simulated surface air temperature (ºC), (d) simulated rainfall (mm·day$^{-1}$), (e) the
differences between observed and simulated surface air temperature (ºC; simulation minus observation) and (f)
the differences between observed and simulated rainfall (mm·day$^{-1}$; simulation minus observation) over the Loess
Plateau in 2001. The observed surface air temperature and rainfall are from the gridded observation dataset
developed by the National Meteorological Information Centre of the China Meteorological Administration. The
simulated surface air temperature and rainfall are obtained by averaging the 11 members (with different initial
conditions) of LCENS$_{2001}$.

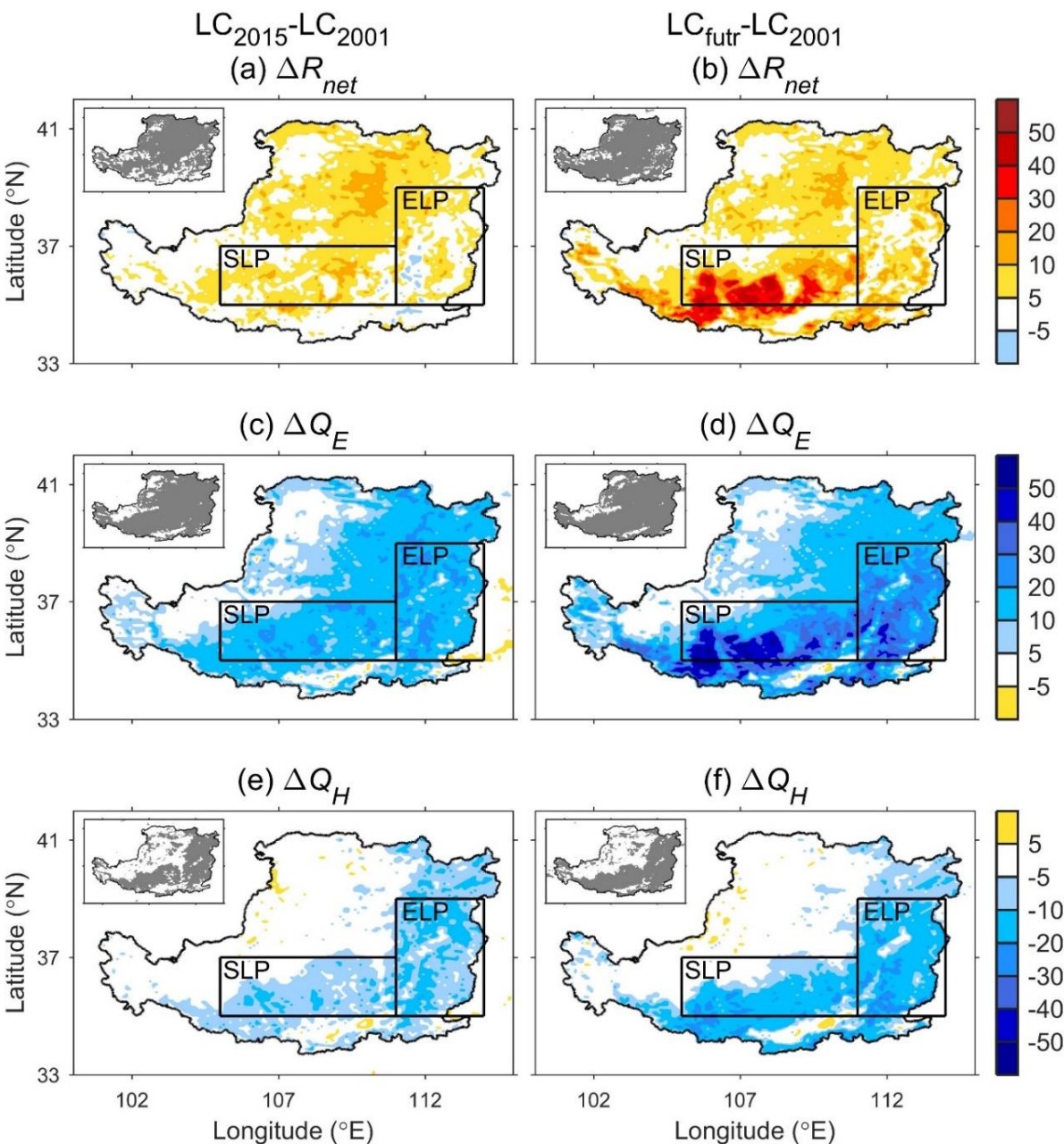


**Figure 5.** Changes in June-July-August-September mean (a and b) land surface net radiation (W·m$^{-2}$), (c and d)

latent heat flux (W·m$^{-2}$) and (e and f) sensible heat flux (W·m$^{-2}$) between the LC$_{2001}$ and LC$_{2015}$ (LC$_{2015}$-LC$_{2001}$; a,

c, and e), and between the LC$_{2001}$ and LC$_{futr}$ (LC$_{futr}$-LC$_{2001}$; b, d, and f) over the Loess Plateau from 1996 to 2015.

The south (SLP) and east (ELP) Loess Plateau regions are defined in Figure 2. The map of statistical significance

test is shown in the embedded figure on the upper left corner of each panel. The grey denotes the local change is

statistically significant at 95% confidence level using a two-tailed Student's *t*-test.

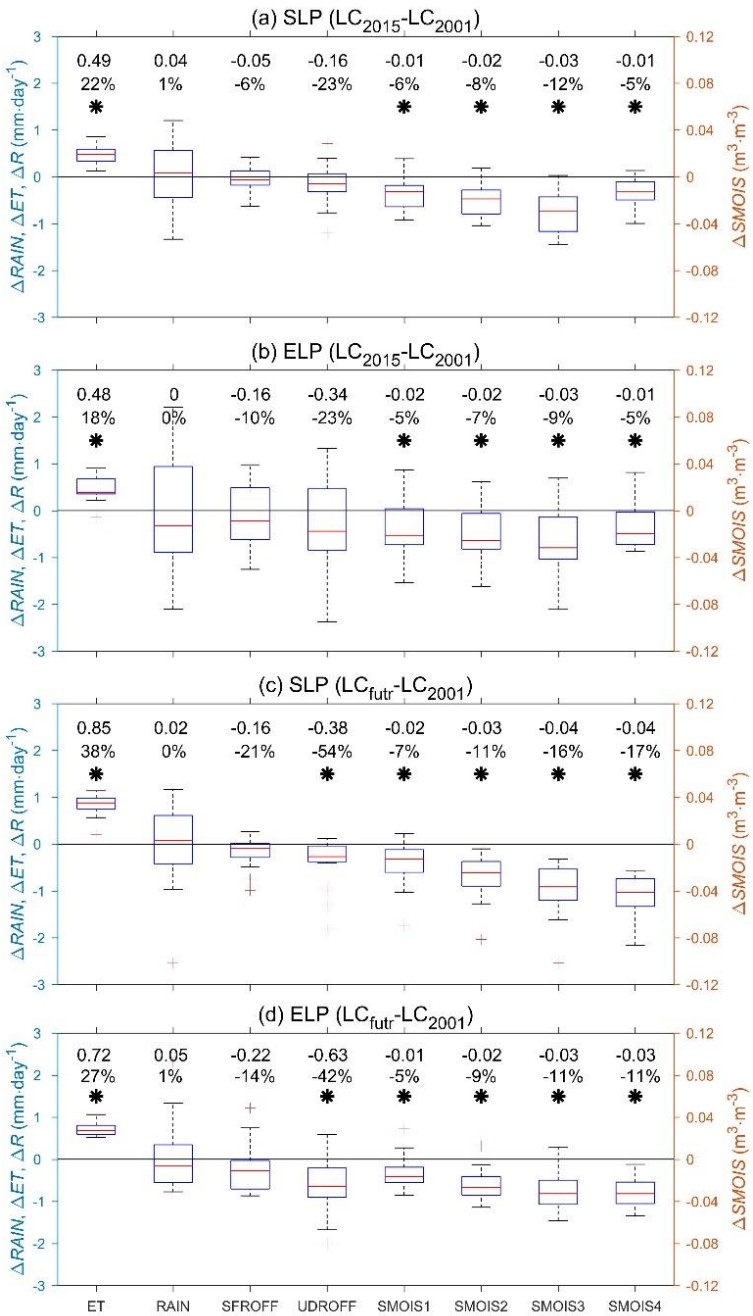

760

**Figure 6.** Box plot of changes in June-July-August-September mean evapotranspiration (*ET*, mm·day⁻¹), rainfall
(RAIN, mm·day⁻¹), surface runoff (SFROFF, mm·day⁻¹), underground runoff (UDROFF, mm·day⁻¹) and soil
moisture (m³·m⁻³) of 1st layer (*SMOIS1*, 0-10 cm), 2nd layer (*SMOIS2*, 10-40 cm), 3rd layer (*SMOIS3*, 40-100 cm)
and 4th layer (*SMOIS4*, 100-200 cm) averaged over (a and c) south Loess Plateau and (b and d) east Loess Plateau
between LC$_{2001}$ and LC$_{2015}$ (LC$_{2015}$-LC$_{2001}$; a and b), and between LC$_{2001}$ and LC$_{futr}$ (LC$_{futr}$-LC$_{2001}$; c and d) from
1996 to 2015. The south (SLP) and east (ELP) Loess Plateau regions are defined in Figure 2. The 1st and 2nd line
members denote absolute and relative changes averaged by twenty members. The black asterisk denotes the
change is statistically significant at 95% confidence level using a two-tailed Student's *t*-test.

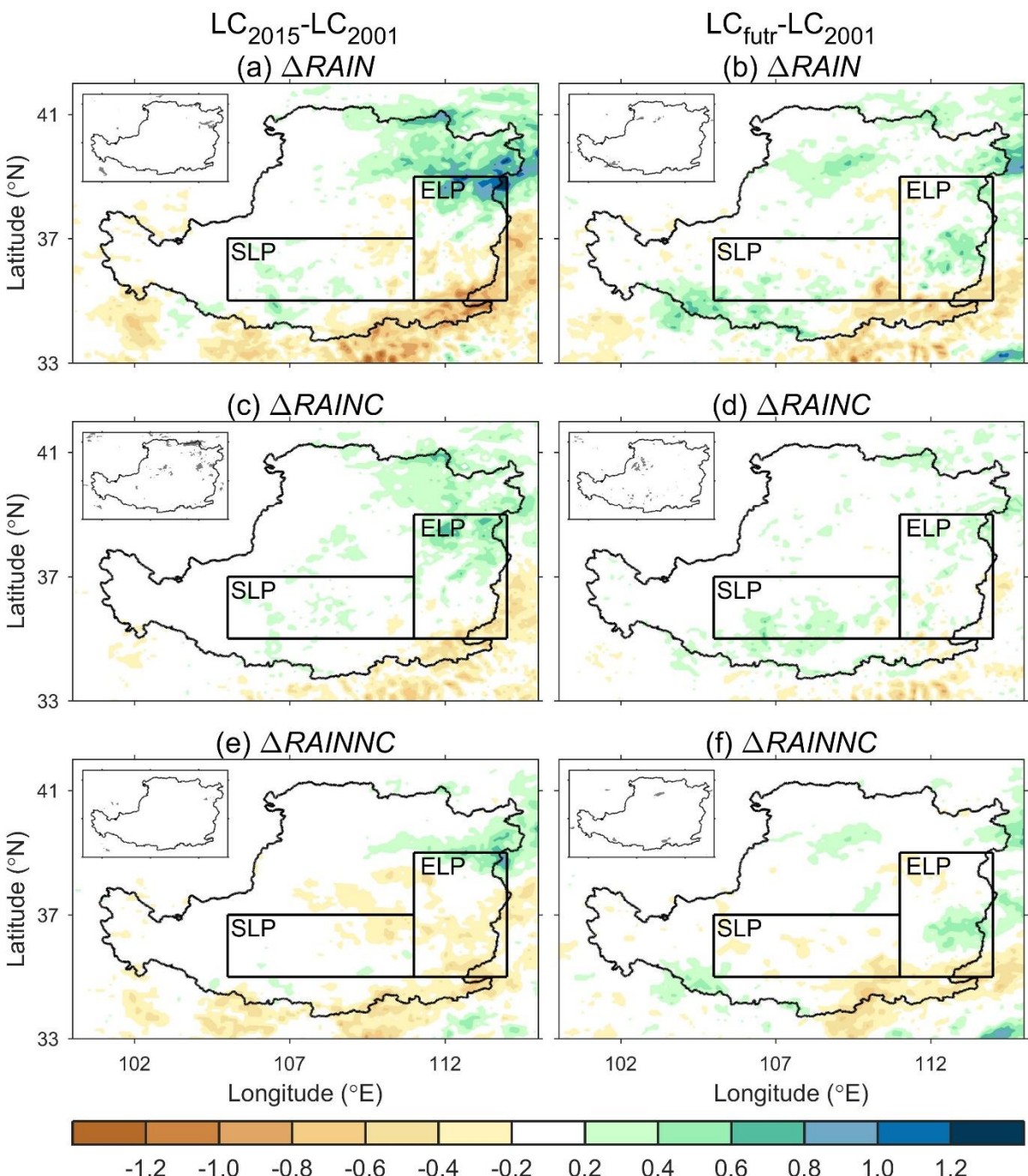

**Figure 7.** Same as Figure 5, but for (a and b) total rainfall (mm·day⁻¹), (c and d) convective rainfall (mm·day⁻¹) and (e and f) non-convective rainfall (mm·day⁻¹). The south (SLP) and east Loess Plateau (ELP) regions are defined in Figure 2.

773

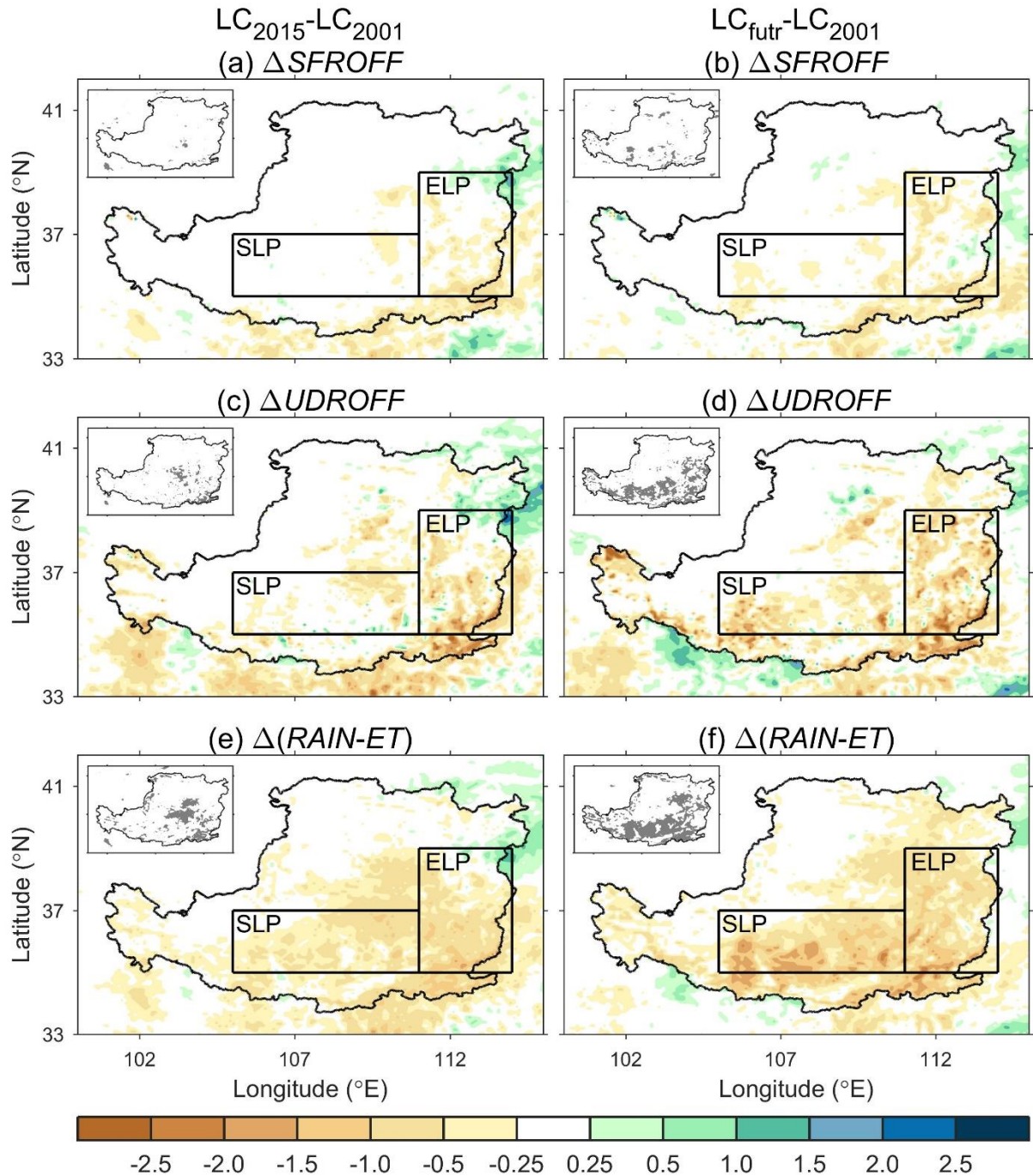

**Figure 8.** Same as Figure 5, but for (a and b) surface runoff (mm·day$^{-1}$), (c and d) underground runoff (mm·day$^{-1}$) and (e and f) rainfall minus evapotranspiration (mm·day$^{-1}$). The south (SLP) and east Loess Plateau (ELP) regions are defined in Figure 2.

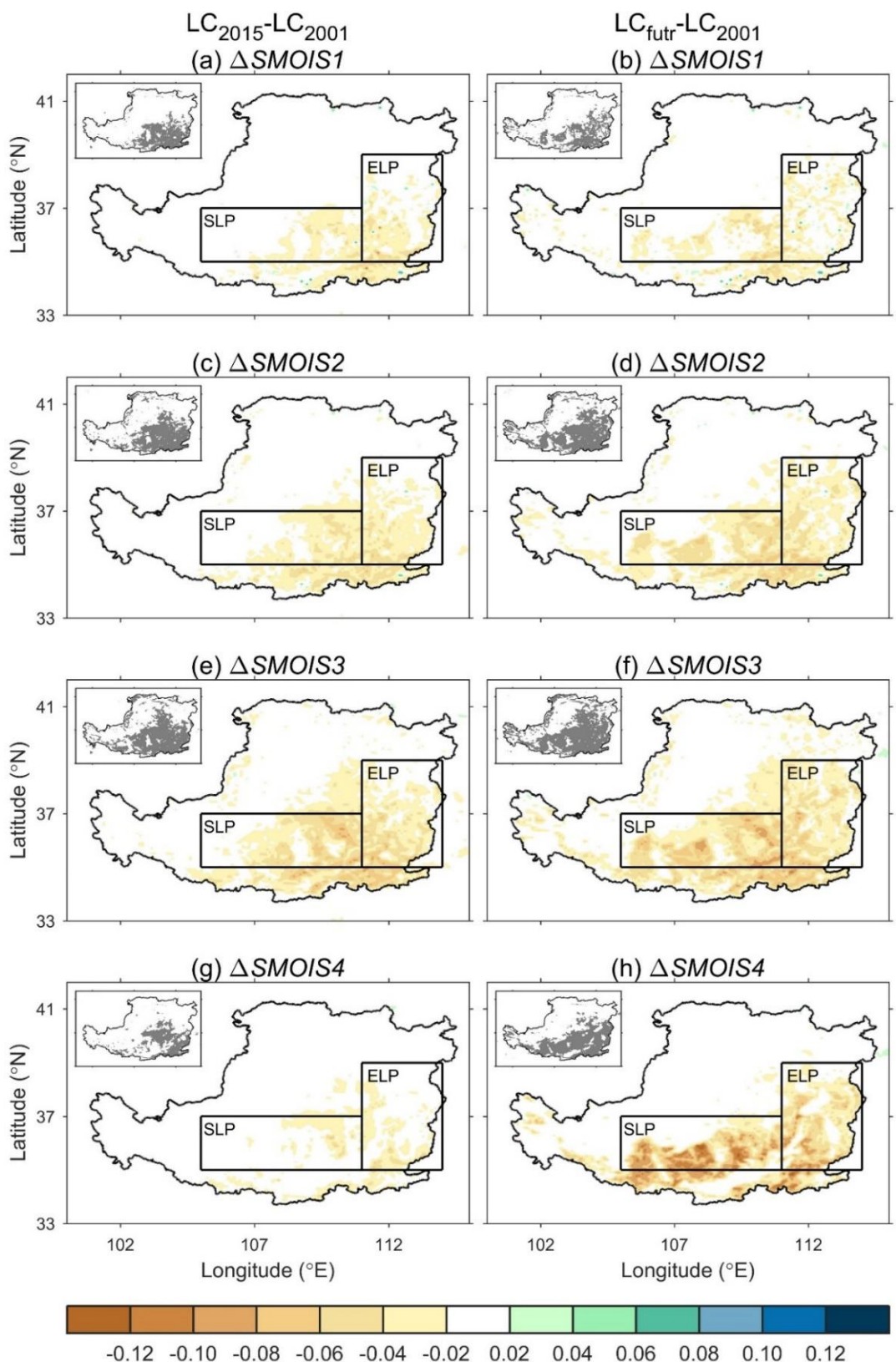

778

**Figure 9.** Same as Figure 5, but for the soil moisture change (m³·m⁻³) of (a and b) first layer (0-10 cm), (c and d)

second layer (10-40 cm), (e and f) third layer (40-100 cm) and (g and h) forth layer (100-200 cm). The south (SLP)

and east (ELP) Loess Plateau regions are defined in Figure 2.

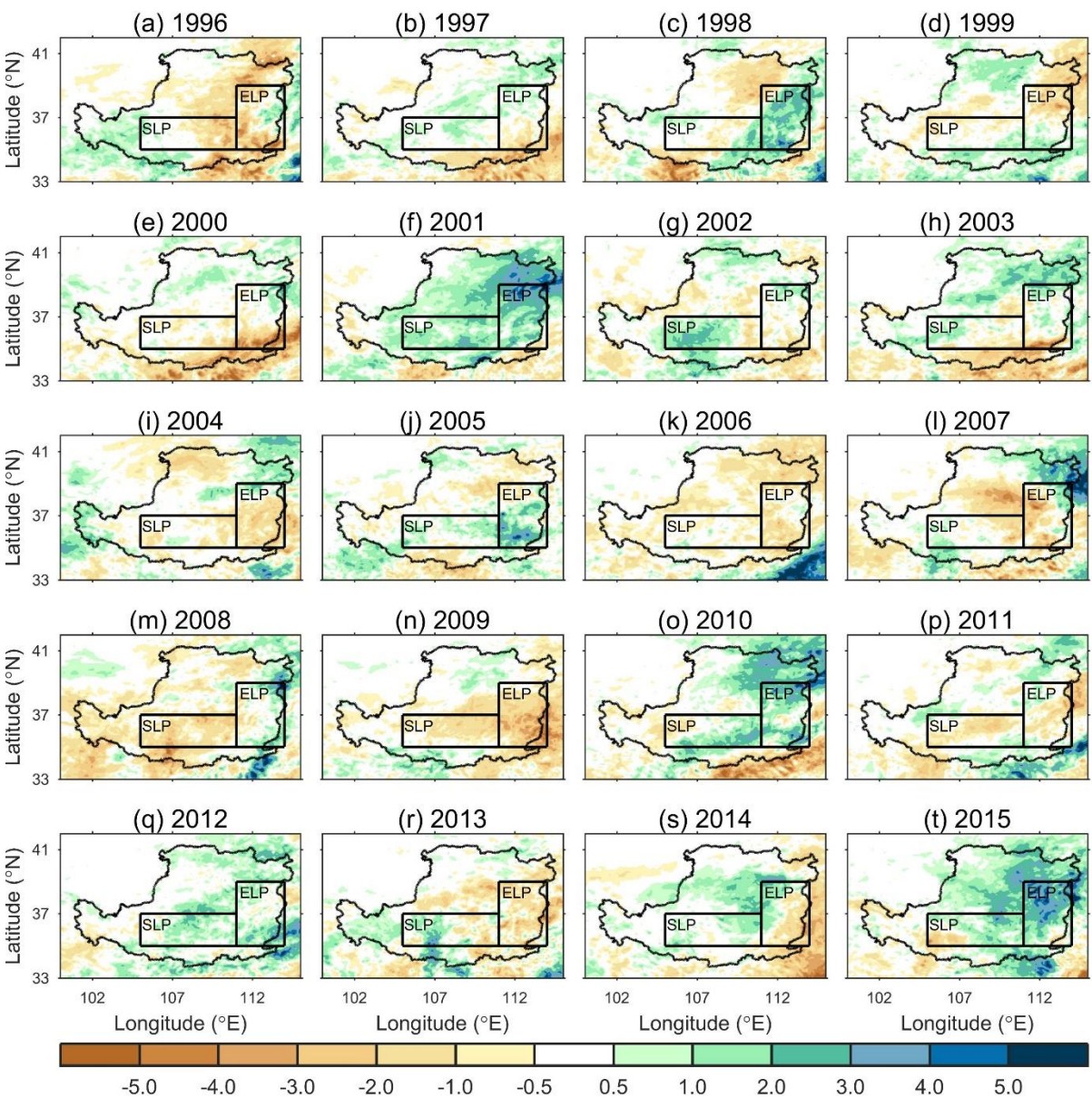

**Figure 10.** Changes in June-July-August-September mean rainfall (mm·day$^{-1}$) of each realisation members (years) between the LC$_{2001}$ and LC$_{2015}$ (LC$_{2015}$-LC$_{2001}$) over the Loess Plateau from 1996 to 2015. The south (SLP) and east Loess Plateau (ELP) regions are defined in Figure 2.

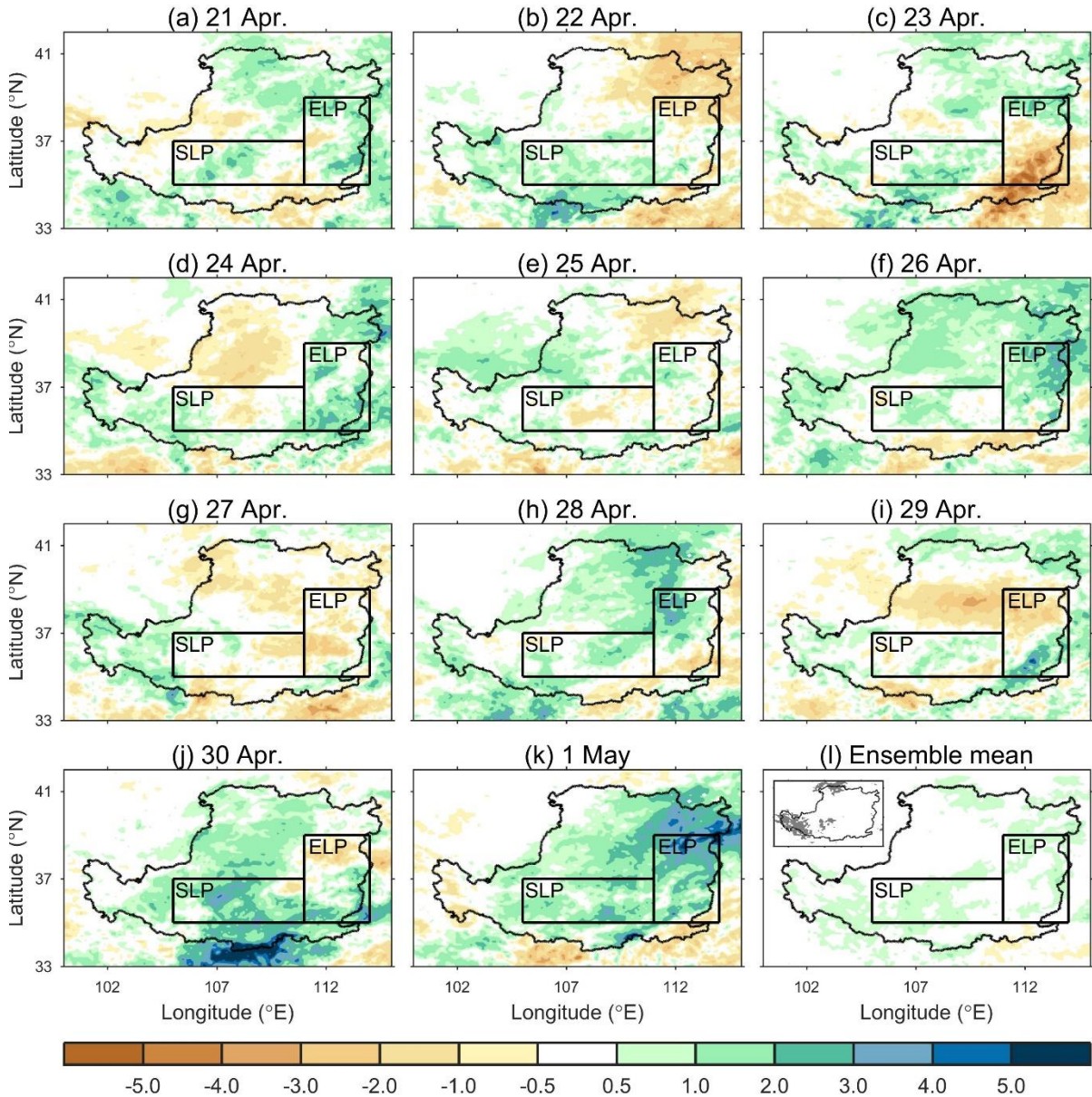

**Figure 11.** Changes in June-July-August-September mean rainfall (mm·day$^{-1}$) of each realisation member (a-k) and ensemble mean (l) between the LCENS$_{2001}$ and LCENS$_{2015}$ (LC$_{2015}$-LC$_{2001}$) over the Loess Plateau in 2001. The south (SLP) and east Loess Plateau (ELP) regions are defined in Figure 2. The map of statistical significance test is shown in the imbed figure on the upper left corner of panel l. The grey denotes the local change is statistically significant at 95% confidence level using a two-tailed Student's *t*-test.

792

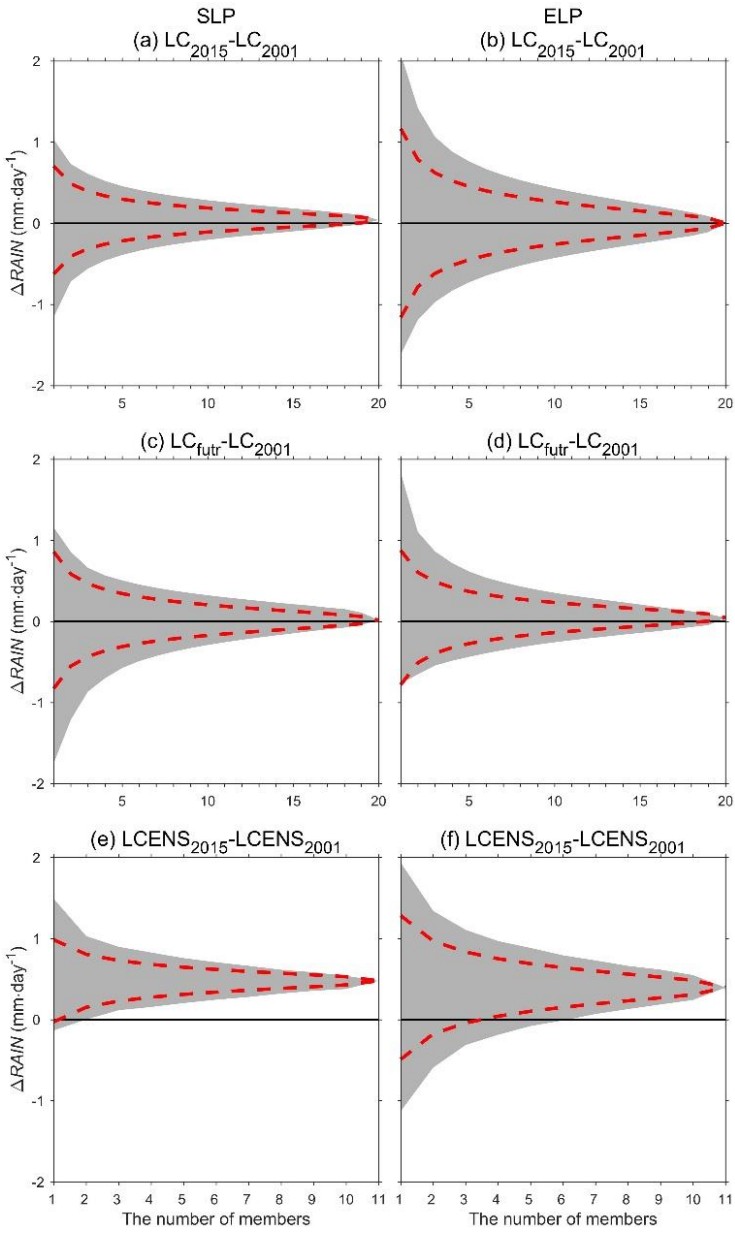

793

**Figure 12.** The relationship between the changes in June-July-August-September mean rainfall (mm·day$^{-1}$) and the number of members. The number of members ranges from 1 to 20 for (a and b) LC$_{2015}$-LC$_{2001}$ and (c and d) LC$_{futr}$-LC$_{2001}$, and from 1 to 11 for (e and f) LCENS$_{2015}$-LCENS$_{2001}$. The mean rainfall change is averaged over (a, c and e) south Loess Plateau and (b, d and f) east Loess Plateau respectively. The south (SLP) and east (ELP) Loess Plateau regions are defined in Figure 2. For a given number of realisations, the rainfall is averaged over these members. The grey area denotes the range of rainfall changes from all possible combinations of a given number of members. The red dashed line denotes the 5$^{th}$ and 95$^{th}$ percentile of the rainfall changes from all possible combination of a given number of members.