# Peer review of "Impact of revegetation of the Loess Plateau of China on the regional growing season water balance"

_Hydrology and Earth System Sciences, 2019_

## Referee Comment (RC1) · Anonymous Referee #1 · 30 Aug 2019

The manuscript by Ge et al. 2019 (HESSD) presents a very important study on exploring the impacts of revegetation of on regional water balance over the Loess Plateau, China. The GFGP was initiated in late 1990s and has tremendous influences on the Loess Plateau. The impacts of revegetation on the region's hydrological balance should be carefully investigated using both observation and model simulations. The current study has tried to answer this question using WRF model, and provide knowledge and information for policy makers. In general, I think it is a very interesting and important study, and I would recommend it for publication after minor revision. Please find my detailed comments below.

1. In introduction section, please add some introduction or examples about published studies that used WRF for hydrological balance analysis. The related references will

[Figure]

add confidence of using WRF in the current study and demonstrate the solidness of the current results.

2. In regard to the findings of the current study, it would be better to add some discussions on similarity/difference with existing studies. It will add extra values if published study has found similar trend in evapotranspiration based on satellite products/ground-based measurements, or model simulations.
* * *

---

## Referee Comment (RC2) · Anonymous Referee #2 · 6 Sep 2019

Review comments for the manuscript "Impact of revegetation of the Loess Plateau of China on the regional growing season water balance" by Jun Ge, Andrew J. Pitman, Weidong Guo, Beilei Zan, Congbin Fu.

General

The paper investigated the impact of revegetation on the hydrology of the Loess Plateau. The introduction needs to be further clarified. For example, the authors stated that "the response of rainfall to large-scale revegetation is rarely investigated". As far as I known, there are studies (e.g., Ma et al., 2013; Chen et al., 2016; Yosef et al., 2018) that have investigated it. Furthermore, the authors mentioned in the discussion section that "Our results are broadly consistent with both field (Jia et al., 2017; Jian et al., 2015; Jin et al., 2011) and satellite (Feng et al., 2017; Li et al., 2016; Xiao, 2014)

[Figure]

observations". Therefore, the new findings in this work need to be further highlighted. The applied land cover change in 2015 relative to 2001 was not consistent with the expected fact. Explanations were missing in several places in the manuscript which kind of focused more on the phenomenon. Detailed comments are given below.

Specific Concerns/Comments

1) Line 55: The authors stated that "the response of rainfall to large-scale revegetation is rarely investigated". As far as I known, there are studies that have investigated it.

Ma, D., M. Notaro, Z. Liu, G. Chen, and Y. Liu. Simulated impacts of afforestation in East China monsoon region as modulated by ocean variability, Climate Dynamics, 41(9-10), 2439-2450, 2013, doi: 10.1007/s00382-012-1592-9

Chen, L., Z. Ma, R. Mahmood, T. Zhao, Z. Li, and Y. Li. Recent land cover changes and sensitivity of the model simulations to various land cover datasets for China, Meteorology and Atmospheric Physics, 129(4), 395-408, 2016, doi:10.1007/s00703-016-0478-5

Yosef, G., R. Walko, R. Avisar, F. Tatarinov, E. Rotenberg, and D. Yakir. Large-scale semi-arid afforestation can enhance precipitation and carbon sequestration potential. Scientific Reports, 8(1), 996, 2018. doi:10.1038/s41598-018-19265-6

Wang, Y. L, Feng, J. M, Gao, H. Numerical simulation of the impact of land cover change on regional climate in China. Theoretical & Applied Climatology, 2014, 115(1-2):141-152

Chen, H. S et al. Numerical Simulation of the Impact of Land Use/Land Cover Change over China on Regional Climates during the Last 20 Years. Chinese Journal of Atmospheric Sciences, 2015

Xu, L., G. Yang, Y. Feng, Y. Du, and X. Han. A study on microclimate impacts of artifical vegetation on the Loess Plateau, Research of Soil and Water Conservation, 17(4), 170-179, 2010

Ma, Y. Climatic and agricultral effect of converting farmland into forest or grass land in ShanGanNing region in China, Chinese Academy of Meterological Sciences and Nanjing University of Information Science & Technology, 2011

2) Lines 69–70: "Thus, the impact of revegetation on the hydrology of the Loess Plateau remains unclear due to the uncertainty in the rainfall response." The conclusion is kind of arbitrary because there are multiple factors, for example, whether the applied land use change data can reflect the reality, and whether a continuous change in the vegetation boundary condition is considered in the modeling. To my knowledge, the existing modeling studies are mainly about sensitivity experiments which cannot exactly reveal what happened in the real world. This manuscript was also a sensitivity experiment. On the other hand, the change in soil moisture under the GFGP was associated with the investigated soil layer depths. The soil moisture above 1 m on the Loess Plateau was mainly controlled by precipitation.

3) As shown in Fig. 2g, the croplands mainly increased from 2015 to 2001, which is contrary to the expected fact. The applied land cover change data cannot reflect the reality well. Consequently, readers may wonder how much the simulation can represent the fact.

4) Lines 70–71: "Moreover, as far as we know, there has been no study investigating how the regional hydrology would be affected if further revegetation was undertaken." The current effects of land use/cover change still need to be further identified. Readers may want to know to what extent can we trust the conclusion of the study with further revegetation?

5) Lines 107–109: "The MCD12Q1 data were reprojected to Geographic Grid data with a resolution of 30 second (approximately 0.9 km) by the MODIS Reprojection Tool to make them suitable for WRF." Why didn't you resample the MCD12Q1 data into 10 km that is exactly the same to the domain 2?

6) Lines 155–156: "We note land cover change here, rather than revegetation or af-
forestation, for two reasons. First, actual land cover changes since the launch of the GFGP are highly spatially heterogeneous." However, the authors mentioned "revegetation" throughout the manuscript including the title and abstract. If the applied land cover change cannot represent the fact, the simulated conclusions cannot provide too much guidance for the implement of GFGP.

7) The significant changes (grey) in rainfall were not located in the main area of vegetation changing under the "Grain for Green Program" in Fig. 11l. What is the reason?

8) Lines 161–168: The used VEGFRA, LAI and $\alpha$ changes also incorporated other factors including improved agricultural management, climate variability, rising atmospheric CO2 concentration and nitrogen deposition. This may interfere the isolation of vegetation change effect. Please clarify.

9) As shown in Fig. 5c, the latent heat flux (ET) increased significantly almost across the Loess Plateau. However, the LAI and land cover almost didn't change in the region except the ELP and SLP (i.e., the region near the internally draining area). Moreover, the extent of changes in the green vegetation fraction was much larger than that of LAI (LC2015-LC2001). Please clarify the reasons. Additionally, what induced the changes in albedo in the region near the internally draining area? Furthermore, the LAI changed in the ELP and SLP regions, but there was almost no change in albedo. What is the reason? The Lines 248-250 need to be further explained.

10) The rainfall change mainly occurred in the region above the ELP (Fig. 7a), which was not consistent with the mainly occurring area of GFGP. What is the reason? Moreover, the convective rainfall increased and non-convective rainfall decreased for LC2015-LC2001 in Fig. 7. Please clarify the reason.

11) Lines 260–262: "Moreover, the increased rainfall in northeast Loess Plateau occurring in LC2015-LC2001 dissipate when further revegetation is implemented suggesting that this change is largely associated with internal model variability." However, the initial conditions were the same between LC2015-LC2001 and LCfutr-LC2001 with the only

differences in land cover and the biogeophysical parameters.

12) Suggest the authors to add one more figure of spatial P-ET changes which is highly correlated with runoff and soul moisture above 1 m.

13) The rainfall responses were obviously different in different years in Figure 10 under the same vegetation change. Please give some explanation. The Figure 12 was used to demonstrate the impact of model internal variability, but one important factor for the phenomenon may be the large variability in rainfall.

14) Lines 365–371: Generally, if a continuous simulation is conducted, much time will be taken. This may be why the simulation periods were usually not too long in a certain number of studies. If long time spans are considered, continuous simulations usually cannot be realized like this study (only including the growing season). On the other hand, the effects of land cover change are likely associated with the backgrounds of circulation, which suggests that the effects could be different for different research time periods. Tobella et al. (2014) reported that tree planting had both negative and positive effects on water resources in drylands and the net effect was the result of a balance between them. Similarly, this manuscript found that there existed both positive and negative effects of vegetation change on rainfall, and the effects were not small as stated in Line 267 and Fig. 6b. The authors concluded that the results show no impact on rainfall in most places of this manuscript. It seems that the expression is inappropriate because the negative and positive effects likely canceled each other out from 1996 to 2015.

15) Lines 307–316: It was first stated that "the RAIN increase in 2001 with an initial date of 1st May is likely associated with internal variability rather than land cover change." Then, it was concluded "the multiyear averaged RAIN change over northeast Loess Plateau for LC2015-LC2001 (Fig. 7a) cannot be robustly linked with land cover change." If the rainfall response was not associated with the land cover change, does it mean all the results in the manuscript were not linked with the vegetation change?

16) The authors mentioned that their simulations were at high resolution of 10 km many times in the manuscript. However, I don't think a 10 km resolution is high nowadays.

17) Suggest the authors to give the study periods (1996-2015 or just 2001) in the figure captions.

18) Typo mistakes:

Line 20: Results suggests that...

Line 122: "As we only focus the growing season" should be "focus on the growing season".

Line 308: "cf. Fig. 7a and 10f"

---

## Author Comment (AC1) · 24 Sep 2019

Response to Anonymous Referee #1

We thank the referee for their positive comments on our manuscript. The referee raised two suggestions:

1. Add some literature in the introduction focused on hydrological balance studies using WRF. We are happy to edit the introduction to add some studies and will do so in a revised manuscript.

2. Add further discussion around similar findings. We will explore the literature to see if we can find suitable and relevant studies and add these into the manuscript.

[Figure]

We therefore expect to be able to revise the manuscript to fully accommodate the comments by Reviewer 1.

---

## Author Comment (AC2) · 24 Sep 2019

Response to Anonymous Referee #2

We thank the referee for their careful and thorough review of our manuscript. We expect to be able to revise the manuscript thoroughly to accommodate their suggestions. We provide an indication of how we will do this below.

Many of these edits will be clarification of our goals, the introduction and particularly the discussion. We do not expect our conclusions to change, but we do suspect there will be some additional caveats to ensure our conclusions cannot be misinterpreted.

Response to General comments

[Figure]

We will modify the introduction as suggested by the reviewer to clarify our paper and add further to the discussion to clarify our new findings.

Response to Specific comments

1. We thank the reviewer for drawing our attention to some literature that we should have cited. We will modify the paper to accommodate some of these. We note some are in the Chinese literature and only available in Chinese – we will likely not incorporate these as this literature is not accessible to the readership of this journal.

2. The reviewer highlights issues with the rainfall response. We will clarify the text to resolve this comment. We agree models cannot fully explain the observed response, but equally observations are always limited in spatial and temporal extent. We will discuss this in a little more detail.

3. The data we use are based on MODIS and are therefore likely quite reliable even if they are not what we might expect.

4. This is a legitimate comment by the reviewer and we will comment on this in the revised paper.

5. We wanted to keep the same spatial resolution as WRF because we only changed the land cover in the Loess Plateau while other regions retained the default land cover. We will clarify this in the revised paper.

6. We will revise this text to clarify and ensure our conclusions are properly in context.

7. This is a demonstration that the signal, from revegetation, is small relative to the internal variability exhibited by WRF. We will clarify this in the revised paper.

8. This is a necessary caveat and could be an interesting area of future work. We can accommodate a brief discussion in the revised manuscript.

9. We will explain this in the revised manuscript as it is not straightforward. We thank the reviewer for highlighting this.

10. Similar to comment #7, this is a demonstration that the signal, from revegetation, is small relative to the internal variability exhibited by WRF. We will clarify this in the revised paper.

11. Again, this is illustrating internal model variability in WRF under circumstances that the signal from the vegetation change is small. We will clarify this in the revised paper.

12. We will explore this and determine whether the additional figure adds value to our analysis and respond appropriately in the revised paper and responses to the reviewers.

13. We think this question is answered by Figures 10 and 11 but we acknowledge that the text does not fully explain this. We will clarify this in the revised paper.

14. Thank-you for drawing our attention to the Tobella et al. paper. The transferability of this paper to our study is not clear; it is an observational paper focused on Africa in a system where the forests and termites need to be considered together. However, the results of the paper do suggest a different sensitivity to those we find for the Loess Plateau and so we will try to incorporate this type of study into a brief discussion. The issue of whether there is an impact when averaged over several years positive and negative changes combine to no impact is obviously a question of timescale and the demand made on water resources. This is a legitimate comment by the reviewer and we will discuss it in the revised manuscript.

15. As with comment #9 this is not straightforward and we will revise the paper to ensure there is no confusion, and we properly discuss the implications of our results.

We thank the reviewer for the minor comments and will resolve those in the revised manuscript.

---

## Author Response (AR1)

**Response to reviewers**

We received two referees' reports. Referees 1 recommended minor revisions, and Referee 2 provided major comments. We address each point made by the referees below. Our responses are in red, and the new text added to the manuscript is indented and italicized.

**Anonymous Referee #1**

The manuscript by Ge et al. 2019 (HESSD) presents a very important study on exploring the impacts of revegetation of on regional water balance over the Loess Plateau, China. The GFGP was initiated in late 1990s and has tremendous influences on the Loess Plateau. The impacts of revegetation on the region's hydrological balance should be carefully investigated using both observation and model simulations. The current study has tried to answer this question using WRF model, and provide knowledge and information for policy makers. In general, I think it is a very interesting and important study, and I would recommend it for publication after minor revision.

Thank you!

Please find my detailed comments below.

1. In introduction section, please add some introduction or examples about published studies that used WRF for hydrological balance analysis. The related references will add confidence of using WRF in the current study and demonstrate the solidness of the current results.

We have added some further literature using WRF for hydrological balance analysis in the revised manuscript:

Line 102: *WRF has been shown to perform well in dynamic downscaling of regional*

*climate over China (e.g., He et al., 2017; Sato and Xue, 2013; Yu et al., 2015).*

*Additionally, WRF has been used to study the impact of land use and land cover change*

*on the hydrological balance at regional scales (Deng et al., 2015; Zhang et al., 2018).*

*While WRF is therefore potentially suitable for evaluating the impact of revegetation on*

*the hydrology of the Loess Plateau we undertake an evaluation of WRF in simulating*

*surface air temperature and rainfall for this region (See Section 3.1).*

2. In regard to the findings of the current study, it would be better to add some discussions on similarity/difference with existing studies. It will add extra values if published study has found similar trend in evapotranspiration based on satellite products/groundbased measurements, or model simulations.

We have added some discussions on similarity/difference with existing studies in the revised manuscript. For example:

Line 383: *Our results on changes of evapotranspiration, soil moisture and runoff are*

*broadly consistent with both field (Jia et al., 2017; Jian et al., 2015; Jin et al., 2011)*

*and satellite (Feng et al., 2017; Li et al., 2016; Xiao, 2014) observations. For example,*

*the spatial pattern of our simulated soil moisture decline in the growing season is*

*similar to observations from the Advanced Microwave Scanning Radiometer on the*

*Earth Observing System by the Japanese Aerospace Exploration Agency (Feng et al.,*

*2017). Although the increased evapotranspiration due to revegetation of the Loess*

*Plateau has been examined before (e.g., Cao et al., 2017, 2019; Li et al., 2018; Lv et al.,*

*2019), the reduction in runoff and soil moisture in response to revegetation of the Loess*

*Plateau, which is consistent with observations, has been rarely reported in modeling*

*results previously. Moreover, our simulated weak response of rainfall to revegetation of*

*the Loess Plateau, which is hard to determine from observations, is useful in assessing*

*the hydrometeorology of this region.*

Line 408: *WRF shows little response of rainfall to revegetation since the launch of the*

*GFGP, which contradicts earlier results (Cao et al., 2017, 2019; Li et al., 2018; Lv et*

*al., 2019).*

Line 443: *These factors account for the discrepancy between our result and another*

*model based study (Li et al., 2018). Li et al (2018) found a positive rainfall feedbacks to*

*greening and consequently small changes in runoff and soil moisture over north China*

*using a Global Climate Model. In contrast, we demonstrate the rainfall change is too*

*small to compensate for the strongly enhanced evapotranspiration, causing a reduction*

*of runoff and soil moisture in response to revegetation over the Loess Plateau.*

**Anonymous Referee #2**

Review comments for the manuscript "Impact of revegetation of the Loess Plateau of China on the regional growing season water balance" by Jun Ge, Andrew J. Pitman, Weidong Guo,

Beilei Zan, Congbin Fu.

General

The paper investigated the impact of revegetation on the hydrology of the Loess Plateau. The introduction needs to be further clarified. For example, the authors stated that "the response of rainfall to large-scale revegetation is rarely investigated". As far as I known, there are studies (e.g., Ma et al., 2013; Chen et al., 2016; Yosef et al., 2018) that have investigated it.

We stated "the response of rainfall to large-scale revegetation is rarely investigated"

following the "The impact of revegetation on evapotranspiration, soil moisture and runoff over the Loess Plateau has been studied" in the original manuscript. Our statement therefore relates to the Loess Plateau. However, we recognize this could have been confusing and therefore we have clarified the introduction:

Line 54: *Despite the increasing observational evidence demonstrating that revegetation*

*tends to impair the hydrological balance of the Loess Plateau, the response of rainfall to*

*revegetation over this region has commonly been overlooked. This is mainly due to the*

*difficulty in detecting the impact of revegetation on rainfall from observations.*

Line 65: *In contrast with observations, modeling can help disentangle the impact of*

*revegetation on rainfall from the impact of other drivers. Cao et al. (2017) and Li et al.*

*(2018) performed numerical experiments over the whole China and demonstrated that the revegetation over the Loess Plateau can enhance the rainfall locally. Very recently, Lv et al. (2019b) and Cao et al. (2019) performed simulations focused on the Loess Plateau to examine the impact of revegetation or afforestation on rainfall. Lv et al. (2019) reported a significant increase in rainfall while Cao et al. (2019) found spatially divergent changes of rainfall. We also note some earlier studies investigating the response of rainfall to land cover change across China (e.g., Chen et al., 2017; Ma et al., 2013; Wang et al., 2014). Unfortunately, these studies either focused less on the Loess Plateau (Ma et al., 2013) or applied land cover changes unable to reflect the revegetation of the Loess Plateau (Chen et al., 2017; Wang et al., 2014). Therefore, large uncertainties remain in the response of rainfall to revegetation of the Loess Plateau owing to inconsistent conclusions derived from limited studies.*

Furthermore, the authors mentioned in the discussion section that "Our results are broadly consistent with both field (Jia et al., 2017; Jian et al., 2015; Jin et al., 2011) and satellite (Feng et al., 2017; Li et al., 2016; Xiao, 2014) observations". Therefore, the new findings in this work need to be further highlighted.

We have further clarified the new findings of this work in the revised manuscript. There are basically three findings of this work. First, we used WRF to demonstrate the reduction in soil moisture and runoff due to revegetation over the Loess Plateau, which has been rarely reported in previous modeling studies and thereby further confirm the observations. We make this clear in the revised manuscript:

Line 383: *Our results on changes of evapotranspiration, soil moisture and runoff are broadly consistent with both field (Jia et al., 2017; Jian et al., 2015; Jin et al., 2011) and satellite (Feng et al., 2017; Li et al., 2016; Xiao, 2014) observations. For example, the spatial pattern of our simulated soil moisture decline in the growing season is similar to observations from the Advanced Microwave Scanning Radiometer on the Earth Observing System by the Japanese Aerospace Exploration Agency (Feng et al., 2017). Although the increased evapotranspiration due to revegetation of the Loess Plateau has been examined before (e.g., Cao et al., 2017, 2019; Li et al., 2018; Lv et al., 2019), the reduction in runoff and soil moisture in response to revegetation of the Loess Plateau, which is consistent with observations, has been rarely reported in modeling results previously. Moreover, our simulated weak response of rainfall to revegetation of the Loess Plateau, which is hard to determine from observations, is useful in assessing the hydrometeorology of this region.*

Second, we used WRF to demonstrate the marginal response of rainfall to revegetation over the Loess Plateau, which contradicts previous modeling results that used older experimental methods. We also demonstrate that the impact of revegetation on rainfall is very likely overestimated in previous studies due to limited members in simulations. As we stated in the revised manuscript:

Line 407: *We focused on the response of rainfall to revegetation over the Loess Plateau,*

*which is probably the most uncertain of the hydrological components. WRF shows little*

*response of rainfall to revegetation since the launch of the GFGP, which contradicts*

*earlier results (Cao et al., 2017, 2019; Li et al., 2018; Lv et al., 2019). Moreover, the*

*rainfall is weakly affected by further revegetation despite large increase in*

*evapotranspiration. We also demonstrate that the rainfall change is strongly affected by*

*internal variability and a large number of realisations are required before any impact of*

*revegetation on rainfall might be robustly identified. We suggest that some previous*

*studies (Cao et al., 2017, 2019; Lv et al., 2019) based on model simulations may have*

*exaggerated the impact of revegetation on rainfall over the Loess Plateau due to the*

*lack of sufficient realisations. For example, Cao et al. (2017, 2019) and Lv et al. (2019)*

*used the same WRF to perform only three or five member simulations, and concluded a*

*significant change in rainfall caused by revegetation over the Loess Plateau. More*

*interestingly, Cao et al. (2017) and Cao et al. (2019) obtained different conclusions on*

*the rainfall change over the Loess Plateau with same WRF model. They used a broadly*

*similar experimental design but different spatial resolution (30 km and 10 km*

*respectively) and simulations from 2001-2002 with three ensembles and consecutive*

*simulation from 2000-2004 respectively. We could also demonstrate large changes in*

*rainfall over the plateau if we chose 3-5 members but we could demonstrate either large*

*increases or large decreases in 3-5 member averages. Returning to Fig. 6, ET shows a*

*highly consistent increase in response to revegetation among the 20 years, suggesting*

*that ET change is robustly linked with revegetation. Although changes in runoff and soil*

*moisture also show large variability among the 20 years, the distribution of the runoff*

*and soil moisture changes are negative biased. More importantly, the distribution of the*

*runoff and soil moisture changes systematically shift towards negative values. This*

*suggest runoff and soil moisture changes are very likely linked with revegetation. The*

*large variability in runoff or soil moisture changes is induced by the large variability of*

*rainfall. Given the tight linkage between rainfall and runoff or soil moisture, the*

*changes in runoff or soil moisture tends to be mistakenly represented if the rainfall*

*change is not robustly examined, and this requires internal model variability to be*

*thoroughly addressed.*

Third, we investigated the potential future impact on the hydrology of the Loess Plateau if revegetation was continued, which has never been assessed before. As we stated in the revised manuscript:

Line 394: *We also investigated the potential future impact on the hydrology of the Loess*

*Plateau if revegetation was continued, which has not been assessed before but is*

*important for both scientific communities and policymakers.*

The applied land cover change in 2015 relative to 2001 was not consistent with the expected fact. Explanations were missing in several places in the manuscript which kind of focused more on the phenomenon.

We obtain the land cover change from the latest version (version 6) of MODIS land cover product, which should be one of the most reliable datasets. We have added some explanations in the revised manuscript accounting for this comment raised by the reviewer:

Line 188: *In addition to the gain of forests (including evergreen needleleaf, evergreen broadleaf, deciduous needleleaf, deciduous broadleaf and mixed forests) and savannas (including woody savannas and savannas), other changes in land cover type include the expansion of croplands (including croplands and cropland/natural vegetation mosaics) at the expense of grasslands and savannas (Fig. 2g). These increased croplands revealed by the MODIS land cover product, which seem unlikely, have been reported previously (Fan et al., 2015; Lv et al., 2019), and are likely associated with expanded irrigation activities along the Yellow River (Fan et al., 2015; Zhai et al., 2015).*

Detailed comments are given below.

Specific Concerns/Comments

1) Line 55: The authors stated that "the response of rainfall to large-scale revegetation is rarely investigated". As far as I known, there are studies that have investigated it.

Ma, D., M. Notaro, Z. Liu, G. Chen, and Y. Liu. Simulated impacts of afforestation in East China monsoon region as modulated by ocean variability, Climate Dynamics, 41(9-10), 2439-2450, 2013, doi: 10.1007/s00382-012-1592-9

Chen, L., Z. Ma, R. Mahmood, T. Zhao, Z. Li, and Y. Li. Recent land cover changes and sensitivity of the model simulations to various land cover datasets for China, Meteorology and Atmospheric Physics, 129(4), 395-408, 2016, doi:10.1007/s00703-016-0478-5

Yosef, G., R. Walko, R. Avisar, F. Tatarinov, E. Rotenberg, and D. Yakir. Large-scale semi- arid afforestation can enhance precipitation and carbon sequestration potential. Scientific

Reports, 8(1), 996, 2018. doi:10.1038/s41598-018-19265-6

Wang, Y. L, Feng, J. M, Gao, H. Numerical simulation of the impact of land cover change on regional climate in China. Theoretical & Applied Climatology, 2014, 115(1-2):141-152

Chen, H. S et al. Numerical Simulation of the Impact of Land Use/Land Cover Change over

China on Regional Climates during the Last 20 Years. Chinese Journal of Atmospheric

Sciences, 2015

Xu, L., G. Yang, Y. Feng, Y. Du, and X. Han. A study on microclimate impacts of artificial vegetation on the Loess Plateau, Research of Soil and Water Conservation, 17(4), 170-179,

2010

Ma, Y. Climatic and agricultural effect of converting farmland into forest or grass land in

ShanGanNing region in China, Chinese Academy of Meteorological Sciences and Nanjing

University of Information Science & Technology, 2011

We thank the reviewer for providing some literature that we should have cited. We think some literature are helpful and related with this work, such as Ma et al. (2013), Chen et al.

(2017), Wang et al. (2014) and Yosef et al. (2018), so we have cited this four literature in the revised manuscript. For example:

Line 70: *We also note some earlier studies investigating the response of rainfall to land*

*cover change across China (e.g., Chen et al., 2017; Ma et al., 2013; Wang et al., 2014).*

*Unfortunately, these studies either focused less on the Loess Plateau (Ma et al., 2013) or*

*applied land cover changes unable to reflect the revegetation of the Loess Plateau*

*(Chen et al., 2017; Wang et al., 2014).*

Line 456: *It is very likely this would be the consequences in some regions such as*

*Amazonia (Lawrence and Vandecar, 2015; Perugini et al., 2017; Spracklen et al., 2018)*

*and Sahel (Kemena et al., 2018; Xue and Shukla, 1996; Yosef et al., 2018).*

We also note the reviewer provided some literature in Chinese, such as Chen et al. (2015), Xu et al. (2010) and Ma et al. (2011). Given that these literature are not accessible to the readership of *Hydrology and Earth System Sciences*, we think it is unsuitable to cite these literature in the revised manuscript.

2) Lines 69–70: "Thus, the impact of revegetation on the hydrology of the Loess Plateau remains unclear due to the uncertainty in the rainfall response." The conclusion is kind of arbitrary because there are multiple factors, for example, whether the applied land use change data can reflect the reality, and whether a continuous change in the vegetation boundary condition is considered in the modeling. To my knowledge, the existing modeling studies are mainly about sensitivity experiments which cannot exactly reveal what happened in the real world. This manuscript was also a sensitivity experiment. On the other hand, the change in soil moisture under the GFGP was associated with the investigated soil layer depths. The soil moisture above 1 m on the Loess Plateau was mainly controlled by precipitation.

We agree and we have rewritten this sentence in the revised manuscript:

Line 79: *Here, we note it might be unfair to directly compare the observational and*

*modeling results because observational results commonly incorporate multiple factors*

*and modeling results are subject to uncertainties in both land cover change and*

*biophysical parametrization schemes implemented in models (de Noblet-Ducoudre et al.*

*2012; Pitman et al. 2009). These intrinsic differences between observational and*

*modeling cannot fully account for the disagreement on the runoff and soil moisture*

*change due to revegetation over the Loess Plateau.*

3) As shown in Fig. 2g, the croplands mainly increased from 2015 to 2001, which is contrary to the expected fact. The applied land cover change data cannot reflect the reality well.

Consequently, readers may wonder how much the simulation can represent the fact.

As we mentioned above, we obtained the land cover change from MODIS land cover product, which is an authoritative dataset worldwide and should be reliable. We have explained why croplands increased from 2001 to 2015 over the Loess Plateau in the revised manuscript:

Line 192: *These increased croplands revealed by the MODIS land cover product, which*

*seem unlikely, have been reported previously (Fan et al., 2015; Lv et al., 2019), and are*

*likely associated with expanded irrigation activities along the Yellow River (Fan et al.,*

*2015; Zhai et al., 2015).*

We have also added some warnings in the revised manuscript. For example:

Line 184: *We note that the difference between LC2001 and LC2015 should not be*

*regarded as equivalent to the impact of GFGP for two reasons.*

Line 449: *Last, we investigated the impact of revegetation or greening, rather than*

*GFGP, on the hydrology of the Loess Plateau. Directly linking our results to the impact*

*of GFGP on the hydrology of the Loess Plateau should be avoided.*

4) Lines 70–71: "Moreover, as far as we know, there has been no study investigating how the regional hydrology would be affected if further revegetation was undertaken." The current effects of land use/cover change still need to be further identified. Readers may want to know to what extent can we trust the conclusion of the study with further revegetation?

We have clarified why we study the impact of further revegetation on the hydrology of the

Loess Plateau in the revised manuscript:

Line 89: *As far as we know, there has been no study investigating how the regional*

*hydrology would be affected by further revegetation over the Loess Plateau, something*

*important for informing policymakers on the mitigation and adaptation of climate*

*change for this region. Additionally, the vegetation over the Loess Plateau is fragile and*

*highly dependent on the water availability (Fu et al. 2017). How the hydrology would be*

*impacted by further revegetation determines the water availability, and in turn how*

*much more revegetation can be sustained over the Loess Plateau. Neglecting this*

*process risks errors in assessing the upper threshold of vegetation of the Loess Plateau*

*(Feng et al., 2016; Zhang et al., 2018). Given the importance of revegetation over the*

*Loess Plateau now and in the future we examine the impact of further revegetation on*

*the hydrology of the Loess Plateau and pay particular attention to the response of*

*rainfall to revegetation.*

5) Lines 107–109: "The MCD12Q1 data were reprojected to Geographic Grid data with a resolution of 30 second (approximately 0.9 km) by the MODIS Reprojection Tool to make them suitable for WRF." Why didn't you resample the MCD12Q1 data into 10km that is exactly the same to the domain 2?

We have further clarified this comment in the revised paper:

Line 134: *We changed the land cover type within the Loess Plateau while retaining the*

*default land cover type for other regions in our experiments (see details in Section 2.3).*

*Therefore, the MCD12Q1 data were reprojected to Geographic Grid data with a*

*resolution of 30 second (approximately 0.9 km) by the MODIS Reprojection Tool to*

*make them consistent with the default land cover map in WRF.*

6) Lines 155–156: "We note land cover change here, rather than revegetation or afforestation, for two reasons. First, actual land cover changes since the launch of the GFGP are highly spatially heterogeneous." However, the authors mentioned "revegetation" throughout the manuscript including the title and abstract. If the applied land cover change cannot represent the fact, the simulated conclusions cannot provide too much guidance for the implement of

GFGP.

We have rewritten this sentence:

Line 184: *We note that the difference between LC2001 and LC2015 should not be*

*regarded as equivalent to the impact of GFGP for two reasons.*

We have also added a caveat in discussions of the revised manuscript:

Line 449: *Last, we investigated the impact of revegetation or greening, rather than*

*GFGP, on the hydrology of the Loess Plateau. Directly linking our results to the impact*

*of GFGP on the hydrology of the Loess Plateau should be avoided.*

We think "revegetation" is a suitable word to basically describe the land cover change of the

Loess Plateau due to a significant greening trend in the past decades. It has been also widely used in previous literature.

7) The significant changes (grey) in rainfall were not located in the main area of vegetation changing under the "Grain for Green Program" in Fig. 11l. What is the reason?

We have demonstrated these rainfall changes (e.g., increased rainfall of the northeast Loess

Plateau) is associated with model internal variability. Please see section 3.6 for details.

8) Lines 161–168: The used VEGFRA, LAI and changes also incorporated other factors including improved agricultural management, climate variability, rising atmospheric CO2

concentration and nitrogen deposition. This may interfere the isolation of vegetation change effect. Please clarify.

We have clarified this comment in the revised manuscript:

Line 449: *Last, we investigated the impact of revegetation or greening, rather than*

*GFGP, on the hydrology of the Loess Plateau. Directly linking our results to the impact*

*of GFGP on the hydrology of the Loess Plateau should be avoided.*

9) As shown in Fig. 5c, the latent heat flux (ET) increased significantly almost across the

Loess Plateau. However, the LAI and land cover almost didn't change in the region except the ELP and SLP (i.e., the region near the internally draining area). Moreover, the extent of changes in the green vegetation fraction was much larger than that of LAI (LC2015-LC2001).

Please clarify the reasons. Additionally, what induced the changes in albedo in the region near the internally draining area? Furthermore, the LAI changed in the ELP and SLP regions, but there was almost no change in albedo. What is the reason? The Lines 248-250 need to be further explained.

We actually masked the small changes in LAI in the original manuscript. We have changed the colorbar scheme of Fig. 3 to avoid readers misunderstanding the biophysical changes. It is visible that LAI indeed change outside ELP and SLP.

The difference between green vegetation fraction change and LAI change is not surprising because green vegetation fraction and LAI are different measures.

To clarify the albedo change, we have added some text in the revised manuscript:

Line 198: *For example, the α decrease mostly occurs over grasslands in northwest (Fig.*

*3e), where land cover type is rarely changed (Fig 2c). This decreased α is attributed to*

*increased precipitation as well as the restoration of grasslands benefiting from the*

*Returning Rangeland to Grassland Program launched in 2003 over this region (Zhai et*

*al., 2015). In contrast, the α change is negligible in the SLP and ELP, owing to the*

*combined effects of increased forests (Fig. 2a) and croplands (Fig. 2d).*

10) The rainfall change mainly occurred in the region above the ELP (Fig. 7a), which was not consistent with the mainly occurring area of GFGP. What is the reason? Moreover, the convective rainfall increased and non-convective rainfall decreased for LC2015-LC2001 in

Fig. 7. Please clarify the reason.

Similar to comment #7, we have demonstrated the increased rainfall of the northeast Loess Plateau is associated with model internal variability. Please see section 3.6 for details.

The reviewer thinks the convective rainfall increased and non-convective rainfall decreased for LC2015-LC2001 in Fig. 7, while these changes are negligible small. Basically the RAIN change is divided almost evenly between RAINC and RAINNC (Fig. 7c and 7e). This demonstrates the weak linkage between RAIN, RAINC and RAINNC changes and revegetation, and these rainfall changes appears randomness which is more likely induced by model internal variability.

11) Lines 260–262: "Moreover, the increased rainfall in northeast Loess Plateau occurring in LC2015-LC2001 dissipate when further revegetation is implemented suggesting that this change is largely associated with internal model variability." However, the initial conditions were the same between LC2015-LC2001 and LCfutr-LC2001 with the only differences in land cover and the biogeophysical parameters.

To clarify this comment, we have rewritten this sentence in the revised manuscript:

Line 297: *Moreover, the increased RAIN in northeast Loess Plateau occurring in LC2015-LC2001 dissipate when further revegetation is implemented while the changes in both land cover type and biophysical parameters are relatively small over this regions. This increased RAIN should be maintained in LCfutr-LC2001 if the change in*

*RAIN is robust for LC2015-LC2001. We will analyse the increased RAIN of the*

*northeast Loess Plateau in LC2015-LC2001in Section 3.6.*

12) Suggest the authors to add one more figure of spatial P-ET changes which is highly correlated with runoff and soul moisture above 1 m.

We have added the figure of spatial P-ET changes in the revised manuscript. Please see Fig.

8e and 8f.

13) The rainfall responses were obviously different in different years in Figure 10 under the same vegetation change. Please give some explanation. The Figure 12 was used to demonstrate the impact of model internal variability, but one important factor for the phenomenon may be the large variability in rainfall.

We have explained why rainfall responses are different in different years:

Line 308: *This large variability in RAIN changes among the twenty members can be*

*attributed to either different boundary conditions (background climate), which causes*

*the impact of land cover change to diverge (Pitman et al., 2011), or model internal*

*variability.*

We further chose 2001 as a case to examine the whether the increased rainfall in 2001 can be robustly linked with revegetation. We demonstrated that the rainfall changes in 2001 are not robust as we can modified the rainfall changes only by changing the initial conditions. As we said:

*Line 357: We therefore show the RAIN change in each realisation for LCENS2015-LCENS2001 in Fig. 11. These eleven ensemble members share the same boundary conditions with small differences in initial conditions. In contrast with the increased RAIN obtained from setting initial date on 1st May (Fig. 10f), the RAIN changes are modified by an advance of 1 to 10 days in initial conditions. For example, WRF cannot simulate the increased RAIN over northeast Loess Plateau when using an initial date of 22nd, 25th, 27th and 30th April, highlighting that the RAIN change is very sensitive to the initial conditions. Thus, the RAIN increase in 2001 with an initial date of 1st May is likely associated with internal variability rather than revegetation. In another words, the RAIN change due to revegetation is negligible relative to the RAIN change induced by internal variability. We therefore conclude that the multiyear averaged RAIN increase over northeast Loess Plateau for LC2015-LC2001 (Fig. 7a) cannot be robustly linked with revegetation.*

Fig. 12 is to demonstrate that running more members and averaging them can effectively reduce the noise induced by model internal variability. Running multiple years or running a year but with multiple realisations are both effective. As a the reviewer said "one important factor for the phenomenon may be the large variability in rainfall", so it is necessary to examine the rainfall change in a single year with multiple realisations, in which case the large variability in rainfall is absent due to the same boundary condition (background climate).

14) Lines 365–371: Generally, if a continuous simulation is conducted, much time will be taken. This may be why the simulation periods were usually not too long in a certain number of studies. If long time spans are considered, continuous simulations usually cannot be realized like this study (only including the growing season). On the other hand, the effects of land cover change are likely associated with the backgrounds of circulation, which suggests that the effects could be different for different research time periods. Tobella et al. (2014)

reported that tree planting had both negative and positive effects on water resources in drylands and the net effect was the result of a balance between them. Similarly, this manuscript found that there existed both positive and negative effects of vegetation change on rainfall, and the effects were not small as stated in Line 267 and Fig. 6b. The authors concluded that the results show no impact on rainfall in most places of this manuscript. It seems that the expression is inappropriate because the negative and positive effects likely canceled each other out from 1996 to 2015.

We agree that the time scale should be considered when the impact of revegetation is evaluated. As we stated in the revised manuscript:

Line 430: *First, observations of soil moisture declines associated with revegetation can*

*be alleviated once trees mature (Jia et al., 2017; Jin et al., 2011). Our simulations only*

*capture an initial decline in runoff and soil moisture linked with the higher*

*evapotranspiration and we note that the impact of revegetation on the long-time trend*

*(25 - 50 years) would be valuable. Second, we used current boundary conditions (1996-*

*2015) for WRF to predict the impact of further revegetation on the hydrology, which*

*means the boundary conditions do not change in the future in response to climate*

*change. This suggests that we might underestimate the impact of further revegetation in*

*the future if future climate of the Loess Plateau suffers from large changes in response*

*to global warming.*

We have added some discussions on the work of Tobella et al. (2014):

Line 459: *Additionally, Tobella et al. (2014) reported a positive impact of trees on soil*

*hydraulic properties influencing groundwater recharging when termite mound is taken*

*into account in Africa. While the termite mound is rare over the Loess Plateau*

*suggesting this positive impact of trees is unlikely to occur.*

The reviewer think "It seems that the expression is inappropriate because the negative and positive effects likely canceled each other out from 1996 to 2015". This statement is true if the rainfall change in individual year can be robustly linked with revegetation. While we demonstrate that the rainfall change in individual year (we take 2001 for instance) cannot be linked with revegetation. In another word, the rainfall change in individual year, which is induced by model internal variability, cancelled each other from 1996 to 2015. It is therefore legitimate to conclude revegetation has no impact on the rainfall of the Loess Plateau.

15) Lines 307–316: It was first stated that "the RAIN increase in 2001 with an initial date of

1st May is likely associated with internal variability rather than land cover change." Then, it was concluded "the multiyear averaged RAIN change over northeast Loess Plateau for

LC2015-LC2001 (Fig. 7a) cannot be robustly linked with land cover change." If the rainfall response was not associated with the land cover change, does it mean all the results in the manuscript were not linked with the vegetation change?

We have added a text to clarify this comment:

Line 421: *Returning to Fig. 6, ET shows a highly consistent increase in response to*

*revegetation among the 20 years, suggesting that ET change is robustly linked with*

*revegetation. Although changes in runoff and soil moisture also show large variability*

*among the 20 years, the distribution of the runoff and soil moisture changes are*

*negative biased. More importantly, the distribution of the runoff and soil moisture*

*changes systematically shift towards negative values. This suggest runoff and soil*

*moisture changes are very likely linked with revegetation. The large variability in runoff*

*or soil moisture changes is induced by the large variability of rainfall. Given the tight*

*linkage between rainfall and runoff or soil moisture, the changes in runoff or soil*

*moisture tends to be mistakenly represented if the rainfall change is not robustly*

*examined, and this requires internal model variability to be thoroughly addressed.*

16) The authors mentioned that their simulations were at high resolution of 10 km many times in the manuscript. However, I don't think a 10 km resolution is high nowadays.

We have replaced "high resolution" with "relatively high resolution" throughout the revised manuscript.

17) Suggest the authors to give the study periods (1996-2015 or just 2001) in the figure captions.

We have added the study periods in the figure captions.

18) Typo mistakes:

Line 20: Results suggests that...

This has been revised.

Line 122: "As we only focus the growing season" should be "focus on the growing season".

This has been revised.

Line 308: "cf. Fig. 7a and 10f"

This has been revised.

[revised manuscript text omitted]